# Identifying the factors governing internal state switches during nonstationary sensory decision-making

**Zeinab Mohammadi** [1] ✉, **Zoe C. Ashwood** [1,2], **The International Brain Laboratory*** & **Jonathan W. Pillow** [1]

Traditional models of perceptual decision-making fail to capture dynamic strategy switching in non-stationary environments, and the factors governing these switches remain unknown. To address this gap, we developed an advanced internal state model with input-driven transitions and observations. Our approach employs a hidden Markov model (HMM) coupled with two sets of per-state generalized linear models (GLMs): a Bernoulli GLM for state- and stimulus-dependent choices, and a multinomial GLM for input-dependent transitions between states. We applied our model to a decision-making task in a non-stationary environment, analyzing hundreds of thousands of trials from a cohort of mice, and found that their behavior can be accurately described by a four-state model. This model identified two engaged states with low biases relative to the stimulus and two disengaged states with pronounced biases relative to the stimulus. Our analyses revealed that mice preferentially used left-bias strategies during left-bias stimulus blocks, and right-bias strategies during right-bias stimulus blocks, achieving high performance even in disengaged states by biasing choices toward the side with greater prior probability. Our model showed that past choices and past stimuli predicted transitions between left- and right-bias states, while past rewards predicted transitions between engaged and disengaged states. In particular, greater past reward predicted transition to disengaged states, suggesting that disengagement may be associated with satiety. Our approach uncovers links between animal behavior, input regressors, and state transitions, highlighting the complexity of adaptive strategies. This provides a foundation for future research in dynamic decision-making models.

Understanding the brain's orchestration of behavior[1–9] and decision-making strategies[10–14] is fundamental to deciphering the complex computations driving adaptive behavior in dynamic environments. Extensive research[11,14–23] has characterized the computational mechanisms governing sensory decision-making across various species and tasks. Bayesian learning and probabilistic modeling techniques have also found practical applications in the study of mammalian and murine behavior[24–30]. Nevertheless, all of these works either assume that animals make decisions using a single, consistent strategy or fail to account for the non-stationary features of their decision-making environments.

A powerful approach for identifying these strategies and their transitions from behavioral time series data involves a hidden Markov

[1]Princeton Neuroscience Institute, Princeton University, Princeton, NJ, USA. [2]Department of Computer Science, Princeton University, Princeton, NJ, USA. *A list of authors and their affiliations appear at the end of the paper. ✉e-mail: zm6112@princeton.edu

model (HMM) with states corresponding to distinct decision-making rules[31–38]. Each of these rules is parameterized by a generalized linear model (GLM), whose weights describe how the animal weighs sensory inputs and other task covariates (such as recent choices and rewards) towards a decision in that state. The resulting modeling framework, known as GLM-HMM, has been used to reveal state-dependent decision-making during courtship behavior in flies[31] and to show that mice rely on distinct neural circuits in different states[34]. Recent work has also revealed lawful relationships between states identified with GLM-HMM and an animal's level of arousal and uninstructed movements[38].

However, these studies have not thoroughly explored the correlation between animal behaviors and input regressors or state transitions. One notable limitation is their challenge in identifying the factors driving state transitions, especially in non-stationary tasks where strategy adjustments are essential as the reward landscape shifts. Even in tasks with stationary rewards, identifying the factors causing an animal to switch from an engaged state to a disengaged or biased state, and vice versa, remains crucial. Standard HMMs, which rely on fixed transition probabilities, fail to capture these dynamics because they assume static probabilities over time. Furthermore, while previous work has modeled time-varying behavior using continuous updates to decision weights, such as PsyTrack[23], these approaches primarily focus on learning dynamics during training, capturing gradual changes in decision strategies. In contrast, our study examines behavior after learning has stabilized, applying a GLM-HMM framework to infer discrete latent states and their transitions. This allows us to capture abrupt strategy shifts rather than continuous within-state learning, providing a complementary perspective on how trained animals dynamically engage with or disengage from the task.

Our research addresses this gap by revealing significant fluctuations in animal behavior across session phases, which allows for the identification of distinct observation and transition strategies in dynamic contexts. Using a multinomial GLM instead of a fixed transition matrix, our approach shows that animal strategies are influenced by internal state changes. This study explores flexible decision-making in a non-stationary environment using a model with two GLMs, incorporating various independent and distinct transition and observation inputs. This model implements flexibility in decision policy by recognizing that the tuning covariates differ across various behavioral contexts and phases.

We applied this GLM-HMM to a mouse decision-making task with time-varying statistics. In this task, the stimulus probability alternated randomly between left-biased and right-biased blocks, with an average block length of 50 trials[39]. We fit models with different numbers of latent states and found that a four-state model provided an accurate yet interpretable description of decision-making behavior. We found that mice employed left-bias strategies during left-bias stimulus blocks and right-bias strategies during right-bias stimulus blocks. They achieved good performance even in disengaged states by favoring the side with greater prior probability. Notably, past choices and stimuli predicted bias state transitions, while past rewards predicted engagement/disengagement transitions, suggesting a link between disengagement and satiety. This study leverages extensive data from tens of mice and hundreds of thousands of trials, ensuring both generalizability and reliability of our findings. Our results contribute significantly to the sensory decision-making field, especially in environments where stable stimulus-action associations are absent. This highlights the need for models that account for rapid internal state shifts and provide insights into the complexity of behavioral dynamics.

## Results
### An internal state model of sensory-decision-making
In this study, we employed the GLM-HMM framework[31–34,40,41], referred to as an input-output HMM, to analyze behavioral data obtained from mice participating in a decision-making experiment. Our model of state-dependent decision-making incorporates two distinct generalized linear models: an observation GLM (GLM-O), which provides the state-conditional probability of the animal's choice on each trial, and a transition GLM (GLM-T), which provides a vector of transition probabilities for the state transitions after each trial. Notably, we specify independent regressor sets for the transition and observation components of the GLM-HMM.

The observation GLMs seek to describe how various task variables (e.g., sensory input, past rewards) affect choice in each state, and thus correspond to distinct decision-making strategies. The probability of a rightward choice in state $k$, based on observation covariates at trial $t$, is given by:

$$p\left(y_t = \text{right} \,|\mathbf{x}_t^{\text{ob}}, z_t = k\right) = \frac{1}{1 + \exp(-\mathbf{x}_t^{\text{ob}} \cdot \mathbf{w}_k^{\text{ob}})} \tag{1}$$

where $\mathbf{x}_t^{\text{ob}}$ denotes the observation GLM covariates, $\mathbf{w}_k^{\text{ob}}$ is the observation GLM weights for state $k$, and $y_t$ reflects the animal's decision.

The transition GLM seeks to describe how task covariates (e.g., elapsed time in session, accumulated reward, previous choices) affect transitions between states. For state transitions, the multinomial GLM predicts the probability of moving to state $k$ at trial $t$, based on transition covariates:

$$p(z_t = k|\mathbf{x}_t^{\text{tr}}) = \frac{\exp(\mathbf{w}_k^{\text{tr}} \cdot \mathbf{x}_t^{\text{tr}})}{\sum_{j=1}^{K} \exp(\mathbf{w}_j^{\text{tr}} \cdot \mathbf{x}_t^{\text{tr}})} \tag{2}$$

highlighting the role of $K$ total states and the transition weights $\mathbf{w}_k^{\text{tr}}$ in modeling transitions based on the trial's context. Here, $\mathbf{x}_t^{\text{tr}}$ denotes the transition GLM regressors. Because we expected the factors governing choices to be different from those governing state transitions, we used different sets of regressors for the observation and transition GLMs (see Methods).

To limit model complexity while still capturing covariate-driven transitions, we used one transition weight vector per destination state ($\mathbf{w}_j^{\text{tr}}$), along with a baseline term for each state pair ($B_{ij}$). Therefore, the probability of transitioning from state $i$ to state $j$ at time $t$ is modeled as:

$$p(z_t = j|z_{t-1} = i, \mathbf{x}_t^{\text{tr}}) \propto \exp\left(B_{ij} + \mathbf{w}_j^{\text{tr}\top}\mathbf{x}_t^{\text{tr}}\right) \tag{3}$$

This structure allows transitions to depend on both the previous state through $B_{ij}$, a learned baseline transition logit from state i to j-and the upcoming state, enabling covariate-driven transitions without requiring separate filters for each origin state. We verified that transitions into a given state did not differ significantly depending on the preceding state, supporting this modeling choice (Supplementary Fig. S11).

### Mice employ different strategies in a non-stationary perceptual task
We analyzed the mice decision-making dataset from the International Brain Lab (IBL)[39]. In this task, mice detect the location of a Gabor patch on the screen and subsequently turn the wheel to the right or left to indicate the stimulus location[42]. Figure 1a shows an illustrative diagram of the IBL sensory decision-making task. Each experimental trial encompasses the presentation of a sinusoidal grating, characterized by gradient values ranging from 0% to 100%. This grating stimulus is selectively presented on either the left or right periphery of the visual display. Subsequently, mice are mandated to discriminate the spatial location of the grating and communicate their decision via rotational manipulation of a wheel, resulting in a left or right turn, which corresponds to the perceived location of the grating stimulus. Successful execution of this task merits a water reward. For further insights into

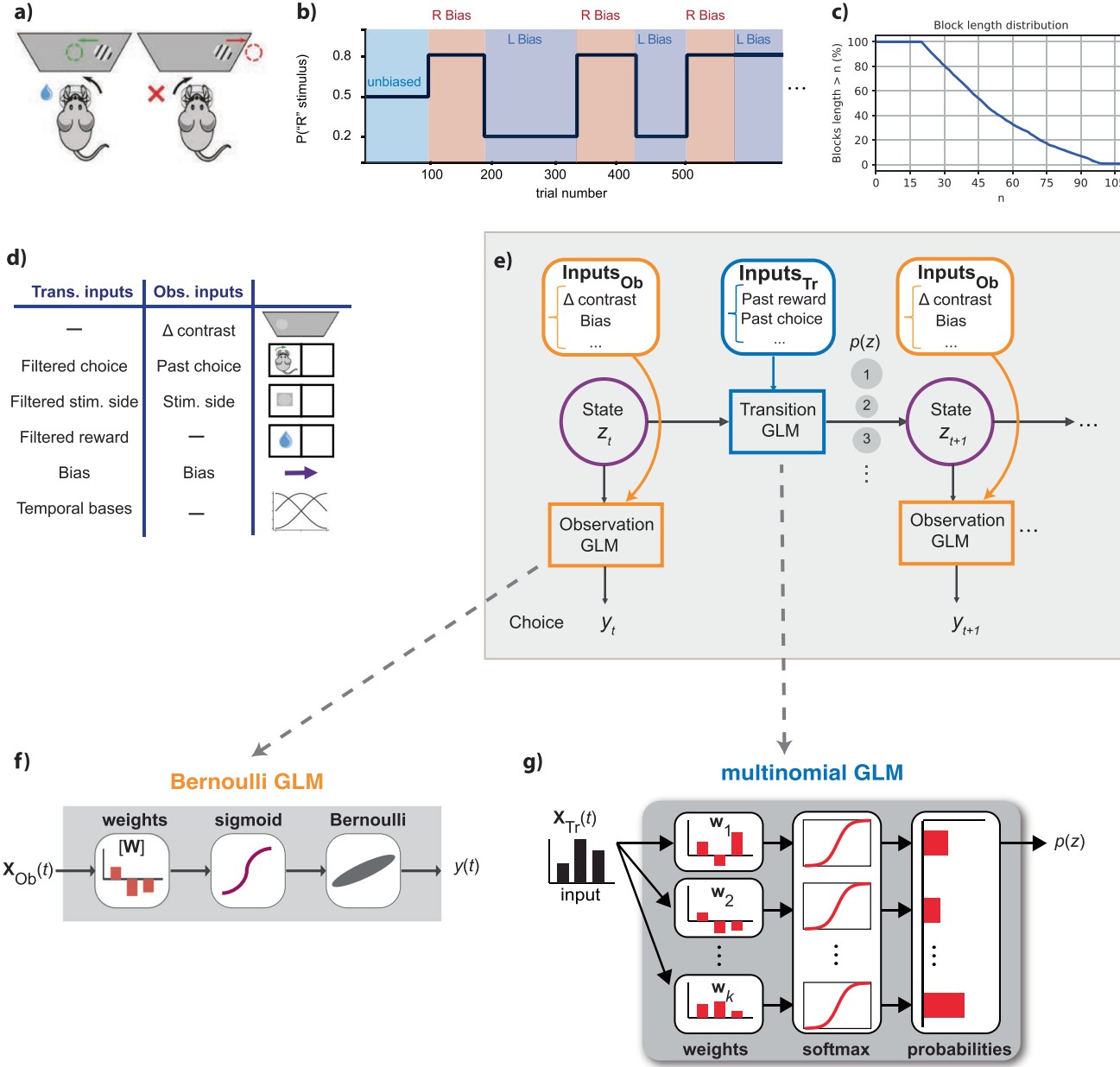

**Fig. 1 | Sensory decision-making task and internal state model (GLM-HMM).**
**a** On each trial, mice were presented with a sinusoidal grating on the left or right side of the screen and were trained to report its side by turning a wheel. Grating contrast was sampled on each trial from a discrete distribution over the values {0, 6.25, 12.5, 25, 50, 100}%. Mice received a water reward for correct responses[39]. **b** Each session started with 90 unbiased trials in which the stimulus had an equal probability of appearing on the left or right. Subsequently, the probability of a right-side stimulus alternated between 0.8 and 0.2 in bias blocks of random duration. Right-bias (R Bias) blocks (red) and left-bias (L Bias) blocks (blue) lasted 50 trials on average, and switches between them were uncued. **c** Distribution of bias-block lengths, which took the form of a shifted, truncated geometric distribution, where n denotes the number of trials (see Methods). **d** We used different sets of inputs to the transition and observation GLMs, reflecting the fact that the

factors governing state switching turned out to be different from the factors affecting choice. (For example, stimulus contrast ($\Delta$ contrast) did not help predict state transitions, nor did past reward predict left-vs-right choice on single trials)[39]. **e** GLM-HMM schematic. The model contains a discrete latent state variable $z_t \in \{1, ..., k\}$, each associated with a distinct Bernoulli observation GLM and a multinomial transitional GLM where $y$ is animal choice. **f** Schematic of Bernoulli GLM, which maps inputs (**X**) to binary choices ($y$) on each trial ($t$), where **W** denotes the GLM weights. **g** Schematic of multinomial GLM, which models the probability of transition to each of $k$ latent states after the current trial. Panels a and d are adapted from Laboratory, T. I. B. et al. Standardized and reproducible measurement of decision-making in mice. Elife 10 (2021). Source data are provided as a Source Data file. L, Left, R, Right, Stim. stimulus, Ob, observation, Tr transition.

this experiment, please refer to the detailed description of the IBL task[39,42].

In this IBL task, after training mice in the foundational purely sensory task, they were introduced to an advanced paradigm where optimal performance necessitated the fusion of sensory perception with recent experience. Specifically, block-wise biases were incorporated into the probability distribution of stimulus locations, thereby

influencing the more probable correct choice. Each session commenced with an unbiased trials block, offering an equal 50:50 probability of left versus right stimulus locations. The length of the unbiased block was 90 trials for all sessions of the task (Fig. 1b). Subsequent trial blocks alternated variably and exhibited biases toward the right and left. The probability distribution skewed at a 20:80 and an 80:20 ratio for the right and left sides, respectively. The length of these

biased blocks ranged from 20 to 100 trials. The transition between these biased blocks was not overtly indicated, necessitating the mice to extrapolate a prior estimation for stimulus location based on recent task statistics. This intricate task compels the mice to assimilate information across multiple trials, strategically employing their prior knowledge to inform their perceptual decisions.

Figure 1c illustrates the distribution of biased data blocks within the IBL experiment, specifically focusing on blocks that extend beyond a defined length n. These extended biased blocks varied in duration, spanning from 20 to 100 trials, with an average length of approximately 58 trials. It's important to note that a typical experimental session consists of several such data blocks.

Within this modeling framework, two distinct sets of inputs were incorporated, encompassing transition and observation inputs as presented in Fig. 1d. In this context, the observed output corresponded to the animal's decision, manifesting as a binary value indicative of the direction in which the wheel was turned-either right or left. A multitude of covariates were additionally integrated into the model and will be elucidated herein. Specifically, the stimulus parameter was defined as the contrast level of a sinusoidal grating, spanning luminance variations between 0% and 100%. For normalization, this stimulus was divided by the standard deviation of trials across sessions. Conversely, the stimulus side parameter characterized the mouse's behavior upon receiving a reward: continued execution of the same choice after reward receipt or a change in behavior in its absence. This binary stimulus side was represented as −1, +1. Furthermore, past choice was established as a binary variable, taking values of −1 or 1, denoting left or right prior choices by the mouse, respectively. The previous reward (pr) value was defined as −1 or 1, corresponding to incorrect or correct decisions, respectively. Moreover, three bases were introduced as linearly independent vectors for the initial 100 trials of each session, capturing the animal's warm-up effect within the model. In the course of this study, the term filtered covariate was employed to reference a covariate subjected to exponential filtering, facilitating the consideration of a temporally filtered variant of the regressor.

Taking into account this description, we included the stimulus, past choice, stimulus side, and bias as regressors for the observation model. Additionally, the covariates influencing GLM transition include a filtered version of the previous choice, previous stimulus, previous reward, and three basis coefficients to capture the warm-up effect. Incorporating an exponential filter for the transition model enables the integration of a temporally filtered regressor. This innovative approach allows for a detailed evaluation of the temporal impact of model regressors on state transitions, enhancing our understanding of the dynamic nature of these transitions.

The GLM-HMM framework used for this analysis is presented in Fig. 1e, which, as mentioned, includes an HMM with a Bernoulli GLM for observations and a multinomial GLM for transitions. We used the IBL data, consisting of extensive data evaluating 123 mice and analyzing all sessions from 37 of them, encompassing hundreds of thousands of trials, to ensure generalizability. From this dataset, we selected 37 mice to fit the GLM-HMM framework with a different number of states. Our animal selection criteria involved including those with a minimum of 30 sessions and incorporating sessions characterized by a low number of error trials, where the animal either did not make a choice or timed out.

In fitting the model, we employed the Maximum A Posteriori (MAP) approach, utilizing the Expectation Maximization (EM) algorithm. The model parameters, denoted as $\Theta = \{\mathbf{w}^{tr}, \mathbf{w}^{ob}, \boldsymbol{\pi}\}$, include the transition weights $\mathbf{w}^{tr}$, observation weights $\mathbf{w}^{ob}$, and the initial state distribution $\boldsymbol{\pi}$. Subsequently, we evaluate the model's log-likelihood based on this approach (see Methods).

## A 4-state model captures decision-making behavior

To identify the optimal model, we fit the GLM-HMM with varying numbers of states and computed the cross-validated log-likelihood on a set of held-out test sessions. After evaluating the log-likelihood values, the GLM-HMM with four states demonstrated notably superior performance compared to the 1-state model. Therefore, we focus on the 4-state model because it comes close to the maximum but is more parsimonious and interpretable than the 5-state model. These four distinct states correspond to four distinct decision-making strategies. The findings suggest that mice consistently employed all states for multiple consecutive trials, with each session seeing the utilization of various states. It was observed that in the GLM-HMM fit of both unbiased and biased data, covariates such as the stimulus (Δ contrast), past choice, stimulus side, and others played crucial roles in predicting the animal's choice.

To conduct this comparison, following the fitting of the model with different numbers of states and computed the log-likelihood of the test data. The EM algorithm was employed for the fitting procedure on the training data sessions. Subsequently, in the testing step, the likelihood of the remaining data was calculated based on the model parameters obtained from the training procedure. By summing over all states, the log-likelihood of the test data is stated as:

$$L = \log\left(\sum_{k=1}^{K} \alpha_{T,k}\right) \qquad (4)$$

in which $\alpha_{T,k}$ is the posterior probability of the mice's decisions from trial 1 to $T$ and was computed solely on the held-out sessions. We can express $\alpha_{T,k}$ as:

$$\alpha_{T,k} = p\left(\mathbf{Y}, z_k | \{\mathbf{X}^{ob}, \mathbf{X}^{tr}\}\right) \qquad (5)$$

where $\mathbf{X}^{ob} = x_1^{ob}, ..., x_T^{ob}$ is the observation input vectors, and $\mathbf{X}^{tr} = x_1^{tr}, ..., x_T^{tr}$ presents the transition input vectors. The animal choices for the specified trials are denoted as $\mathbf{Y} = y_1, ..., y_T$.

First, we performed the global fit in which the model was applied globally to the entire IBL dataset, and the normalized log-likelihood (NLL) for this data was calculated. The defined NLL, measured in bits per trial (bpt), provides a more intuitive understanding of the model's performance and facilitates meaningful comparisons across various models and datasets. This can be expressed as:

$$L_{bpt} = (L_t - L_0)/(T_t \log(2)). \qquad (6)$$

In this equation, $L_0$ represents the log-likelihood of the baseline model, and $T_t$ is the number of trials in the test set. The equation has been divided by $T_t \log(2)$ to present the value per trial. The difference between $L_t$ and $L_0$ represents the enhancement in log-likelihood due to the performance of the GLM-HMM.

Analyzing the Test NLL plot (Fig. 2a), it is evident that the four-state model demonstrates better performance and interpretation, particularly when considering the structured nature of both biased and unbiased data. So, derived GLM weights within the context of the four-state model unveil distinct patterns, which are shown in Fig. 2.

In Fig. 2a, for comparative analysis of different models, we computed the log-likelihood for all animal data. The computation strategy involved 5-fold cross-validation with reserved sessions. In this model, when evaluated on the held-out data, the four-state GLM-HMM performed much better than the GLM itself (1-state model). Additionally, the GLM-HMM successfully captured the temporal pattern of inhibition influencing the animal's decision-making process. Also, a comparison was drawn in terms of the Test Normalized Log Likelihood (NLL) between the model solely incorporating the Bernoulli GLM and the model encompassing both the GLM-O (Observation) and GLM-T (Transition). This is presented in Fig. 2a. We can see similar effects and

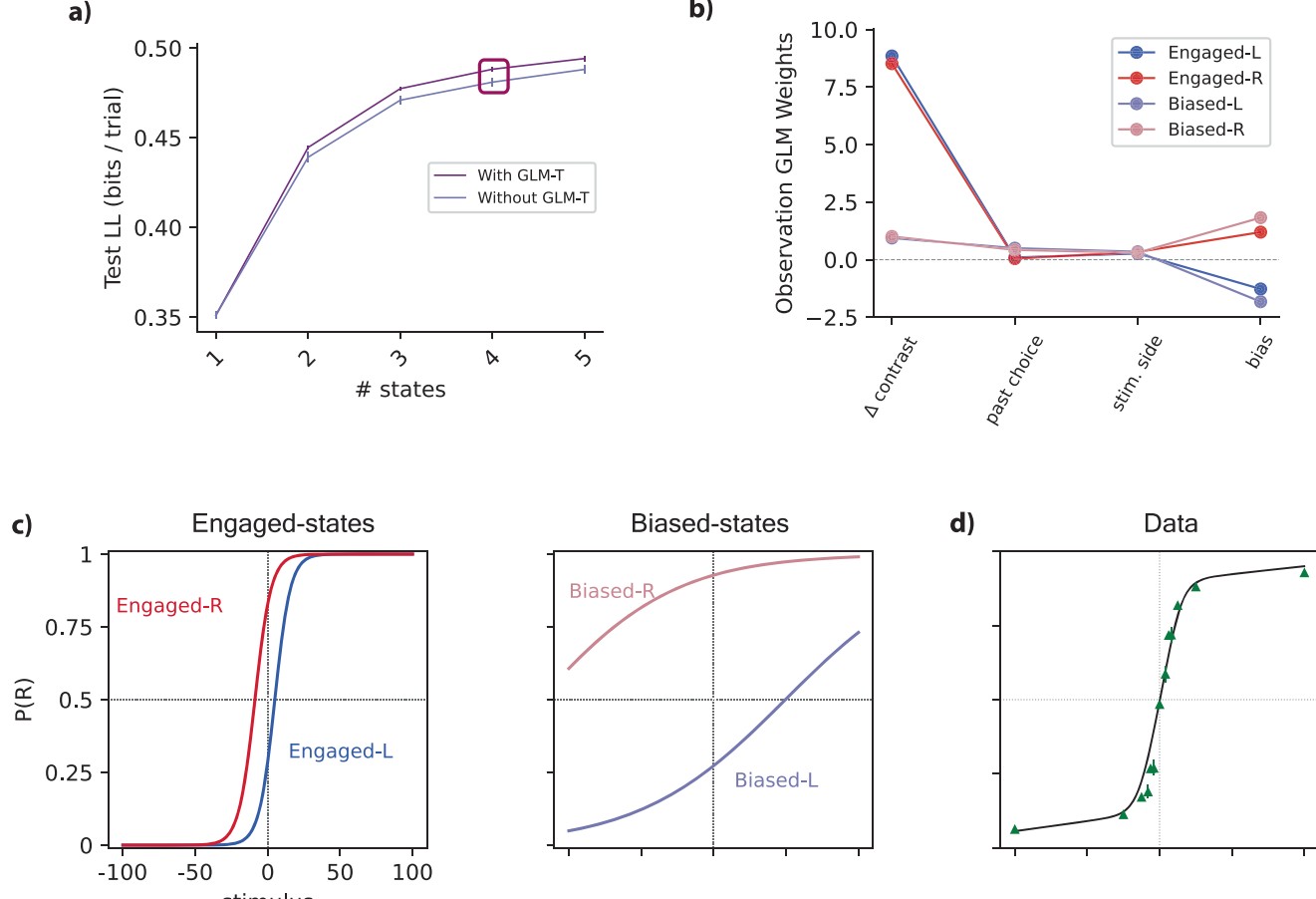

**Fig. 2 | Analysis of GLM-HMM fits. a** Test log-likelihood (LL) of the GLM-HMM with and without multinomial GLM transition, as a function of the number of latent states, using pooled data from all 37 animals in our dataset. The model with GLM transition (GLM-T) outperformed the basic GLM-HMM for all models with multiple states. The four-state model (purple box) came close to saturating the test log-likelihood yet had highly interpretable states, and we therefore selected it for further analysis. Data are presented as mean test log-likelihood; error bars denote 68% bootstrap confidence intervals across cross-validation folds (5-fold cross-validation). **b** The fitted observation-GLM weights for the four-state model revealed two Engaged states, which we refer to as Engaged-L and Engaged-R. In these two states, the weight on stimulus contrast (Δ contrast) was large and the bias weight

was negative or positive, respectively. The other two states exhibited small stimulus weight and a large left or right bias, leading us to refer to them as Biased-L and Biased-R states. These biased or disengaged states also exhibited small weights on previous choice, indicating a greater tendency to preserve. **c** State-specific psychometric curves for each of the four states in the fitted model, which show how the GLM weights in (**b**) map signed stimulus contrast into the probability of a rightward choice. **d** The model-based psychometric curve for an example mouse is a linear combination of the state-specific psychometric curves shown in (**c**), and provides a close match to the empirical choice data for an example mouse (green triangles). Source data are provided as a Source Data file. Stim. stimulus.

improvement for the 5-state model as shown in Supplementary Fig. S1a. Our analysis revealed an enhanced performance of NLL in the presence of the transition GLM (GLM-T) within our model, as opposed to the scenario wherein the transition GLM was absent. Although the difference between the two models with and without GLM-T (Fig. 2a) is not very large, this is because the results are shown in log-likelihood per trial (in bits). Even a very small improvement in log-likelihood per trial can result in a much better model fit (refer to Eq. 32 in the Methods, but in this case comparing models with and without GLM-T instead). For example, for the 4-state model, the model with GLM-T is ~0.015 bits per trial higher than the model without GLM-T. This means that for a dataset with 400 trials, the data would be about $2^6 \approx 64$ times more probable under the GLM-T than without it. For a dataset with 4000 trials, the data would be about $2^{60} \approx 1.15 \times 10^{18}$ times more probable under the GLM-T model. This exponential scaling highlights how even small per-trial improvements translate into dramatically better fits for larger datasets.

Although the NLL plot shows a slight increase for the 5-state model, our focus here remains on the 4-state model for the sake of simplicity and interpretability. This choice is motivated by the fact

that, in the 5-state model, four states closely resemble those in the 4-state model and the new state, named Past-choice, emphasizes the role of past choice. However, our emphasis in this work revolves around the stimulus and bias roles, which are crucial for analyzing biased block data. A comprehensive description of the 5-state model is available in Supplementary Figs. S1 and S2, providing insights into the observation and transition weights, and analysis of all five states.

In Fig. 2b, state 1, Engaged-L, exhibits a substantial weight attributed to the Δ contrast (stimulus) and a moderate weight associated with the left bias. State 2, designated as the Engaged-R state, features a significant weight related to the stimulus and a moderate right-bias weight. Conversely, states 3 and 4, denoted as Biased-L and Biased-R, respectively, exhibit reduced stimulus weights. Nevertheless, bias weights generate a strong left bias for state 3 and a right bias for state 4. Additionally, in these states, a nominal weight is assigned to the previous choice factor.

These state-specific weights influence the shape of the psychometric curves, a fundamental tool in psychophysics[43,44] and decision-making modeling. Typically characterized by a sigmoid-shaped function, the psychometric curve is intricately linked to a linear

representation of the stimulus (Δ contrast) and augmented by a bias term. This mathematical framework is widely adopted to capture the relationship between stimuli and an individual's responses. Here, the psychometric curve graphically represents the choice probability (right side) relative to stimulus contrast. State-specific psychometric curves were meticulously generated within the framework of a four-state GLM-HMM presented in Fig. 2c. The resulting curves serve as intricate depictions of the behavioral responses observed within each state. The psychometric curves for Biased-L and Biased-R states exhibit shallower inclines, indicative of notable leftward and rightward biases, respectively.

In Fig. 2d, the green triangles correspond to the experimental choice data of the mouse (alongside 95% confidence intervals). Also, to derive the solid black line, a temporally sequenced dataset was generated to match the trial count of the example mouse. This process involved using the meticulously fitted parameters of the GLM-HMM specific to this animal and the actual sequence of stimuli presented during its trials. For each trial iteration, the probability of making a rightward choice (p(R)) was calculated for each of the nine potential stimuli, regardless of the actual stimulus presented. This calculation entailed averaging the per-state psychometric curves, as illustrated in Fig. 2d while adjusting their weights according to the pertinent row in the transition matrix, as this adjustment was contingent on the latent state sampled in the preceding trial.

## Modeling the dynamics of state transitions in animal behavior

Figure 3a represents both the stimulus and the animal's choice on the same trial, complemented by the transition regressors of the model. These transition regressors include filtered stimulus side, filtered choice, and filtered reward, providing a comprehensive illustration of the multinomial GLM inputs in the model.

The multinomial GLM weights for transition between states are shown in Fig. 3b. For states 1 and 3, characterized by a left-biased component, negative weights are assigned to the filtered choice and filtered stimulus side. This suggests that these factors contribute to the transition towards left-biased states (Engaged-L or Biased-L). Conversely, for the right-biased states, including states 2 and 4, positive weights are assigned to the filtered choice and filtered stimulus side, indicating a tendency towards right-oriented choices. Additionally, in disengaged states (states 3 and 4), the positive weight of the filtered reward signifies its role in decision-making, as the animal is less attentive to the stimulus in these states. This pattern is reversed in engaged states (states 1 and 2), where the stimulus is pivotal, resulting in a negative weight for the filtered reward. Therefore, past rewards were found to predict transitions to disengaged states, suggesting a connection to satiety. Also, transitions between left- and right-bias states were influenced by past choices and stimuli. Notably, the term filtered covariate here refers to transition regressors subjected to exponential filtering, facilitating the integration of temporally filtered versions of these regressors in the analysis.

On the other hand, Supplementary Figs. S2 and S3 present an analysis of transition weights and transition patterns between different states, as well as the dwell time distribution for the model with 5 states. The calculation of dwell time in each state involves utilizing the diagonal elements of the inferred transition matrix of the 5-state model. This allows for a nuanced examination of the temporal dynamics and the relationships between states, contributing valuable insights into the overall understanding of the 5-state model.

Additionally, we introduced a set of three temporal basis vectors to capture the time dependence of state transitions at the beginning of each session. The temporal modulation of the bases' effect is illustrated in Fig. 3c. These bases were strategically integrated to capture the gradual adaptation of the animal during the initial 100 trials of each session, effectively accounting for the warm-up effect within the model. Furthermore, as depicted in Supplementary Fig. S4, we present

the impact of transition weights and the associated effects of these bases, showing the transition weight corresponding to each basis in a specific state for both the 4-state and 5-state models. This representation provides insights into the influence and weight of each basis in the warm-up effect.

Figure 3e presents the inferred transition matrix pertaining to the four-state GLM-HMM, designed to accommodate the full IBL dataset. Evident within this matrix are notable magnitudes along the diagonal, serving as clear indications of a heightened probability for the system to sustain its presence within the same state during transitions. This prominence underscores the significant propensity for the model to exhibit persistence and stability within individual states. Figure 3d displays the transition bias for pooled IBL data, derived from the intercept terms of the GLM-T model (prior to input modulation). These values represent the model's default expectations about the likelihood of remaining in the same state or transitioning to another, independent of trial history or covariates. Visualizing these biases provides insights into the intrinsic stability of each behavioral state-for example, higher diagonal values indicate stronger self-transitions and more persistent states.

To compare models with different filtering parameters and choose the best one, we compared the normalized test Log-Likelihood (test LL) across models with various numbers of states, varying the values of $\tau$. Here, $\tau$ represents the time constant of the exponential filter applied to the transition covariates and is associated with the duration of the history considered for these covariates. Figure 3f illustrates the normalized test LL for different values of $\tau$ while adjusting the number of states. In Fig. 3g, a similar comparison is shown, but for models with different numbers of states. It is evident that the model with the specific value of $\tau = 4$ yields the best test log-likelihood results, and we have chosen this value for filtering the transition covariates of the model.

## Exploring the temporal patterns within the model

Our model, harnessing the power of the fitted GLM-HMM, captures the temporal dynamics and diverse decision-making strategies of mice, enhancing our understanding of their cognitive processes. The findings regarding the dynamics of the model over time are presented in Figs. 4 and 5. This model allowed us to calculate the posterior probability of the mouse's hidden state throughout all the trials. These state trajectories represent our best estimates of the mouse's internal state on each trial, considering the entire sequence of observed inputs and choices during a session.

The plots showcase the posterior probabilities and correct/incorrect animal choices associated with their corresponding sessions in four distinct states, as presented in Fig. 4a. These visualizations incorporate a background color scheme that serves to distinguish biased blocks: the color pink corresponds to right-biased blocks, while the blue shade denotes left-biased blocks. This color-coded representation enhances the comprehensibility of the data, facilitating the identification of patterns within the biased segments of the experiment. This figure illustrates that states exhibiting right bias, namely Engaged-R and Biased-R, demonstrated a greater likelihood within blocks characterized by a right-biased orientation (designated by a light pink background color). Left-oriented blocks with a blue background hue exhibited a similar observation, where states influenced by left bias exhibited higher probabilities. Also, as we can see Fig. 4a, what emerged is a clear pattern: in most trials, one state stood out as significantly more probable than the others, signifying a strong level of confidence in our understanding of the mouse's internal state based on the observed data.

We made a histogram to elucidate the distribution of the first transition into states aligning with related bias within each data block (Fig. 4b). This analytical approach is geared towards capturing the number of trials spanning from the commencement of a biased data

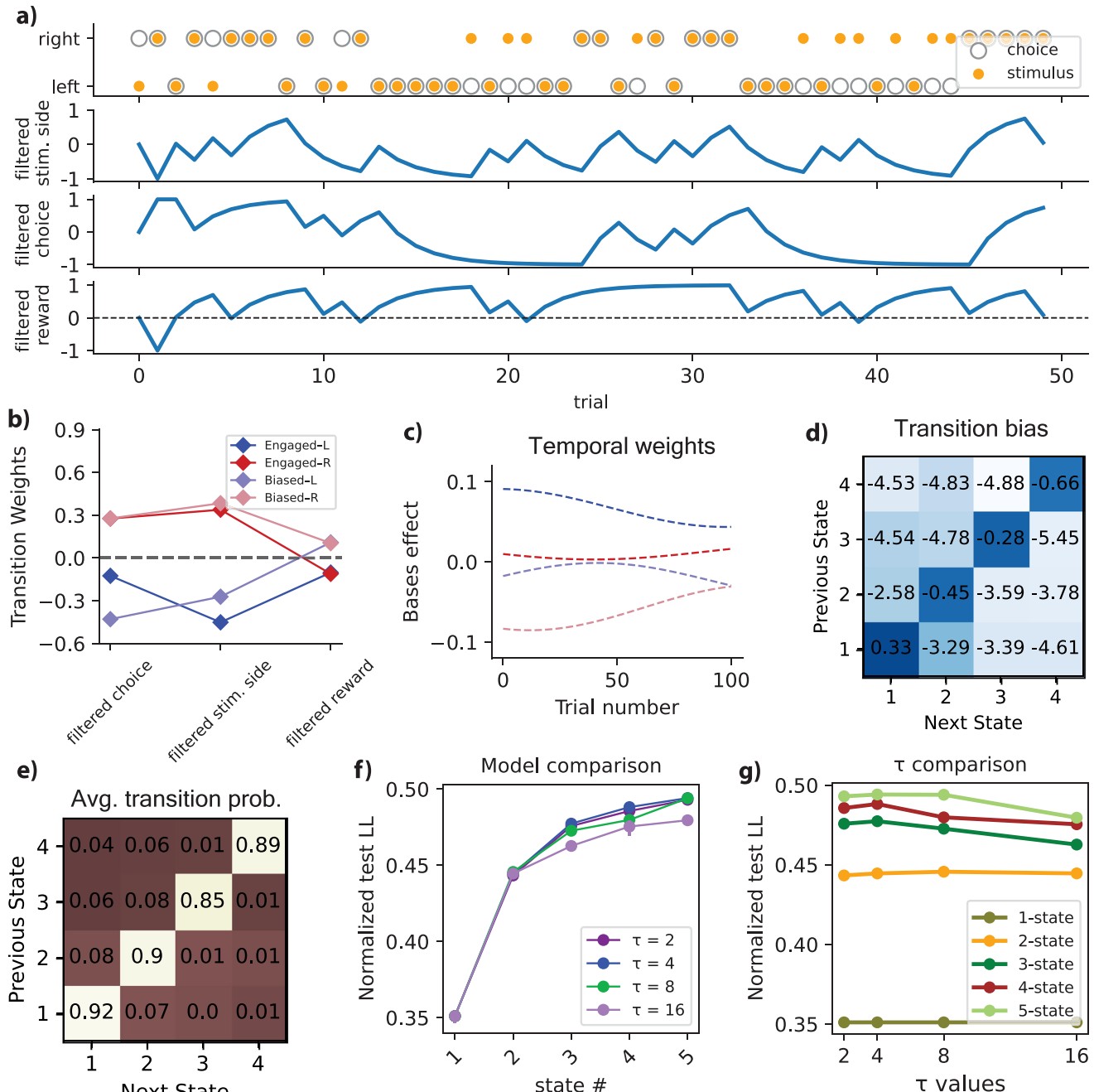

**Fig. 3 | Transitional and temporal dynamics of the model. a** The stimulus (Δ contrast) and animal's choice, along with the transition regressors (filtered stimulus side, filtered choice, and filtered reward), where filled yellow and open circles denote the stimulus and choice, respectively. **b** The transition GLM weights indicate that left-biased states (1 and 3) are associated with negative weights for filtered choices and stimuli, while right-biased states (2 and 4) have positive weights. Also, engaged states (1 and 2), show negative weights for filtered rewards, whereas disengaged states (3 and 4) show positive weights for filtered rewards. This suggests that past choices and stimuli drive transitions between left- and right-bias states, while past rewards are linked to disengagement, potentially indicating satiety. **c** Temporal modulation of the bases effect, which results from the multiplication of bases weights by bases traces (three bases were introduced into the model to capture the animal's warm-up effect during the first 100 trials of each session). **d** The transition bias of the model for pooled IBL data. **e** The deduced transition matrix for the four-state GLM-HMM, tailored to the entirety of mice IBL data. This matrix displays prominent values along the diagonal, indicative of a pronounced likelihood of persisting within the same state. **f**, **g** Comparing different models for various values of the filter time constant (τ) for transition covariates: **f** Comparing models with varying numbers of states while changing the values of τ. Analyses used pooled data from *n* = 37 mice (biological replicates; independent animals from the IBL dataset). For each τ, models were trained and evaluated with fivefold cross-validation (folds at the session level). Data are presented as mean normalized test log-likelihood; error bars denote 95% bootstrap confidence intervals across cross-validation folds (fivefold cross-validation). **g** Plotting normalized test log-likelihood versus τ for different models. The best plot corresponds to τ = 4, and this is the value we selected for our analysis. Source data are provided as a Source Data file. Stim., stimulus; Avg., average.

block until the occurrence of the first transition to the corresponding biased states. For instance, in the context of a right-biased data block, this entails quantifying the number of trials required for transitions to the states characterized by a right bias, namely

Engaged-R, and Biased-R. A similar assessment is made for left-biased data blocks and their respective states, with a high value on the left bias weight. The analysis underscores that the median value of this histogram stands at 9 trials, while the maximum value observed is 6

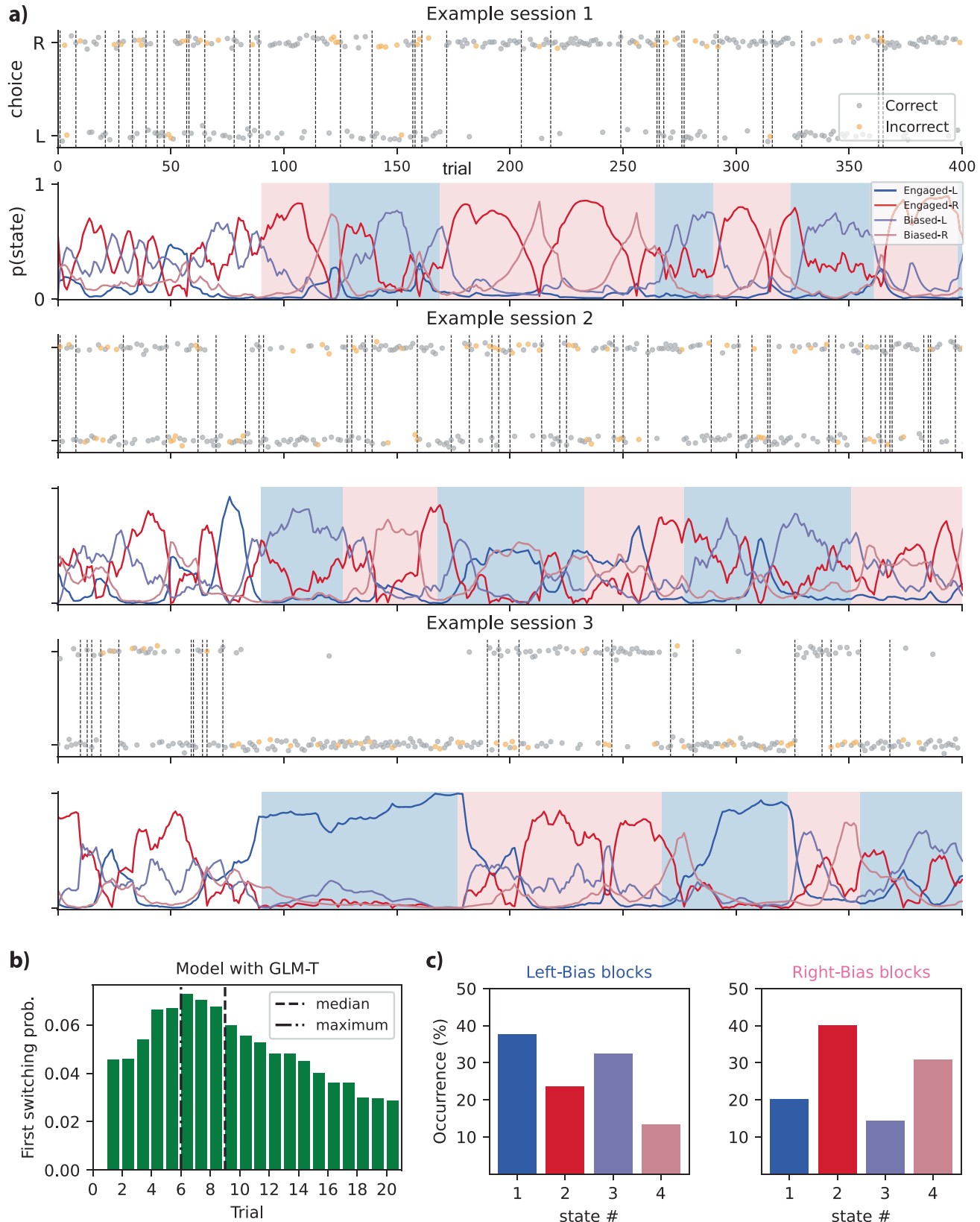

trials. This implies that, on average, it takes approximately 9 trials for mice to transition from the start of a biased data block to a state matched with the prevailing bias.

In Fig. 4c, the fractional occupancy of the four distinct states across the entirety of trials for right-biased and left-biased data blocks was analyzed. Each trial was assigned to the state that exhibited the

highest likelihood, and the proportion of trials designated to each respective state was computed.

As evident from the analysis (derived from data of all 37 mice) for each bias-related plot, the corresponding states were observed to encompass a substantial portion of the entire trial set, indicating a significant representation of the mouse's behavioral responses within

**Fig. 4 | Exploring chronological patterns of animal behavior. a** The plots show posterior probabilities and mice's choices-correct (gray) and incorrect (yellow)-during the task, for four distinct states. Based on the calculated posterior probabilities, the black dashed vertical lines indicate the points where state changes occur. The background color distinguishes biased blocks: pink indicates right-biased blocks, while the blue shade represents left-biased blocks. **b** The histogram for the number of trials of the initial transition probability into corresponding biased data blocks. In the context of right-biased data, the right-side states was generated for each block (e.g., the trial count for initial transition to states with a right bias for right-biased data blocks) **c** An evaluation of the fractional occupancy pertaining to the four discrete states. This was conducted over the entire span of trials encompassing both right-biased and left-biased blocks. The results are presented here separately for the two different data block types. Source data are provided as a Source Data file. Prob. probability;.

biased data blocks. In the context of right-biased data, the right-side plot in Fig. 4c, Engaged-R and Biased-R (represented by red and pink columns) exhibited notably higher values in comparison to the other associated columns.

A similar observation was made for left-biased blocks (left-side plot in Fig. 4c), where Engaged-L and Biased-L showed elevated values compared to the other two states. This observation underscores the impact of bias weights on biased data blocks, highlighting the adaptive nature of the mouse's decision-making process within the experimental framework. Remarkably, the analysis uncovered that the mice spend ~72% of their time in related biased states (e.g., Engaged-R and Biased-R in right-biased blocks), with the engaged state having a slightly higher occurrence chance. In stark contrast, the mice allocated a relatively smaller fraction of their trials, ~28%, to the other unrelated states (e.g., Engaged-R and Biased-R in left-biased blocks).

In the plots of Fig. 5a, we depict the multiplication of transition weights and transition inputs, providing a visual representation of the temporal patterns exhibited by the weighted transition regressors. The specific data for these plots corresponds to the same three sessions of a single mouse as in Fig. 4a. These visualizations offer insights into the nuanced dynamics of the relationship between transition weights and inputs across the given sessions.

Also, a histogram was created to show the frequency distribution of inferred state changes per session across all 58 sessions of data for this mouse in Fig. 5b. In this plot, we considered sessions with lengths more than $T$ trials and analyzed the first $T$ trials of each session. Here, $T$ represents the average duration of all sessions conducted for the respective animal, which was 821 trials. Notably, in about 18% of all sessions, the mouse exhibited fewer than 10 state changes, leading to an average state duration of 154 trials (calculated by dividing 821 trials by 5-state changes). This observation highlights a consistent tendency for this mouse to remain in specific states for extended periods, particularly in high-performance states. These prolonged periods of stability were often interrupted by state changes, primarily occurring when the mouse either adopted a new strategy or shifted its attention to different covariate effects.

On the other hand, in the majority of sessions, approximately 63%, the mouse underwent more dynamic behavior with more than 40 state changes, averaging around 15 trials per state (derived by dividing 821 trials by 55 state changes). This frequent state-switching behavior reveals a decision-making process where the mouse continuously adapted its strategies and attention throughout the session, while still maintaining an average of 15 trials per switch. The notable variation within and between sessions highlights that these different rates of state changes might be due to varying strategies, environmental factors, or animal behaviors on separate days or in different sessions.

Furthermore, our analysis delved into the dynamics of state transitions by utilizing the diagonal components within each transition matrix. These components allowed us to calculate the expected duration of residence, referred to as dwell time, for each animal within distinct states, as illustrated in Fig. 5c. The median duration of residence in the Engaged-L and Engaged-R states was approximately 15 and 10 trials, respectively, while for the Biased-L and Biased-R states, it was 10 and 12 trials. This observation indicates that the animal spent considerable periods in all four states, shedding light on the persistence of specific behavioral states in different strategies.

### Analyzing each individual animal yields consistent findings

To gauge the universality of our findings, we applied the GLM-HMM to the choice data obtained from all animals in the IBL dataset separately (37 individual fits for all mice). As a result, two sets of GLM weights (observation and transition weights) for these animals are presented in Fig. 6a, b. A notable level of substantial agreement was distinctly apparent upon analyzing the fits of the four-state GLM-HMM in the study. This consensus was particularly pronounced, as a significant majority of the mice showcased discernible states, identified as Engaged-L, Engaged-R, Biased-L, and Biased-R (Fig. 6a, b). This alignment in the identification of states underscores the robustness and consistency of the applied GLM-HMM framework in capturing these behavioral patterns across the population of interest for both observation (Bernoulli GLM) and transition weights (multinomial GLM).

The transition and observation weights for the 5-state model, resulting from individual fits for all animals, are thoroughly presented in Supplementary Fig. S5. As we can see, these weights exhibit mostly similar patterns and in the near ranges across the various animals. Notably, this consistency extends to the fifth state, Past-choice, where the observed patterns remain consistent in both observation and transition analyses. This shows valuable insights into the universality of the observed patterns within the studied population.

Figure 6c presents an analysis of the test log-likelihood variation considering different numbers of states for each mouse in the studied population. Each line on the graph represents an individual mouse's data, providing an overview of log-likelihood changes across different state configurations. The solid red line shows the average across all animals, serving as a reference point for comparison. The findings in Fig. 6c are consistent with trends observed across the entire cohort, indicating a strong pattern. Notably, the four-state GLM-HMM consistently outperformed the single-state GLM during cross-validation. This trend was consistent across all 37 mice in our study, reinforcing the reliability and validity of our multi-state GLM-HMM framework as a powerful tool for understanding decision-making behavior in mice across diverse individuals. Similarly, Fig. S6 presents the GLM weights of the 5-state model applied individually to all IBL animals, showing both observation and transition weights. The global fit is shown as a solid black line, with an example mouse fit overlaid as a dashed black line.

### Model performance analysis and comparison

Figure 7 presents a detailed analysis of model performance and a comparison of models with and without GLM-T. Figure 7a shows the difference in $T_{90}$ (90th percentile response time; disengaged - engaged) across mice, demonstrating that the model with GLM-T (purple) achieves higher separation of engaged and disengaged states in terms of response times for all mice, indicating better performance in this aspect. Figure 7b illustrates the average state probabilities across trials, with sessions interpolated to 100 points and then averaged across all sessions and animals. It highlights an initial warm-up effect in which mice start in disengaged states (orange) before transitioning to engaged states (green). Also, this plot suggests a link between disengagement and satiety, as past rewards predict transitions to disengaged states, leading to increased disengagement state occupancy toward the end of the session when mice have accumulated more rewards. Figure 7c further supports this relationship, showing a

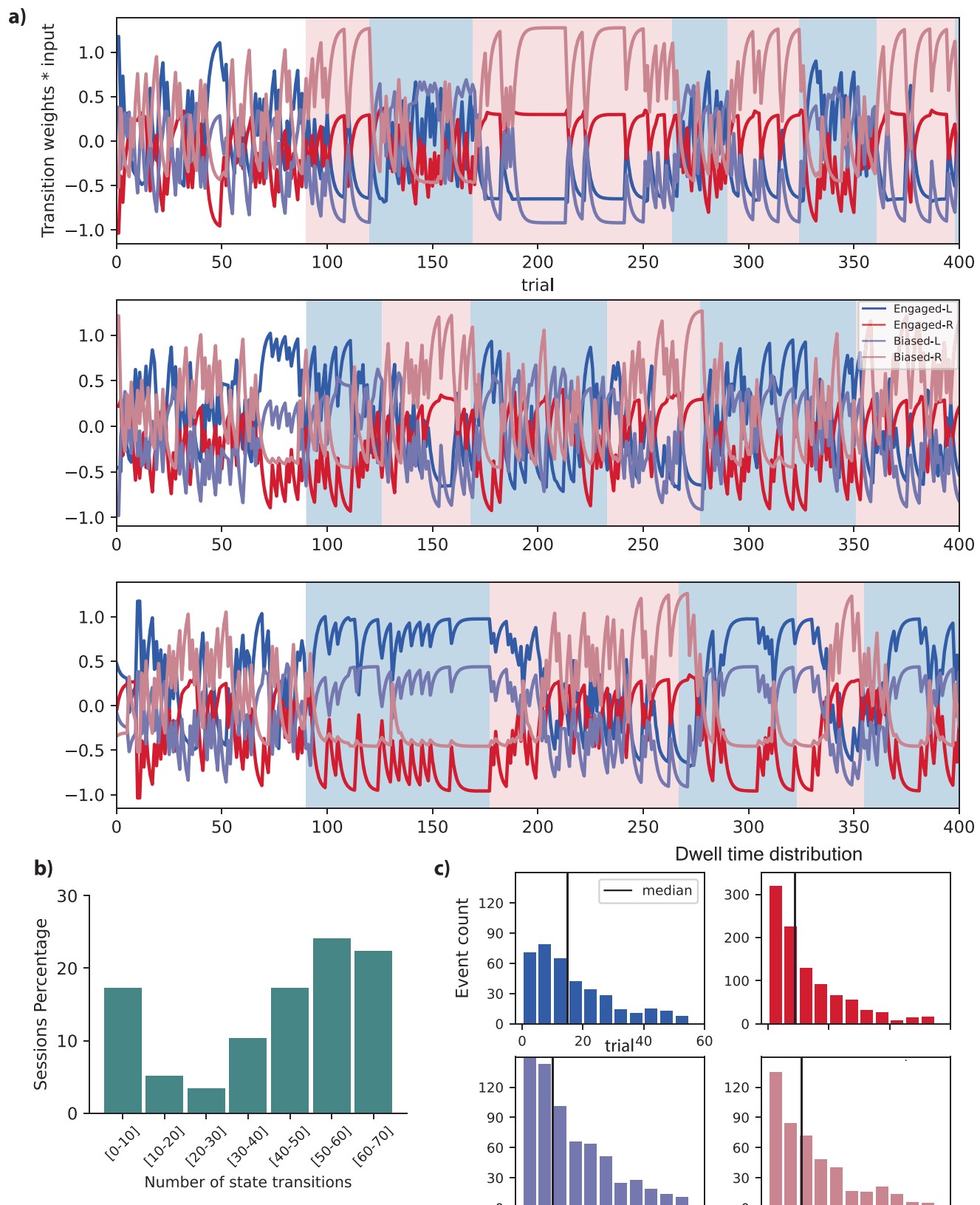

**Fig. 5 | Further exploration into the temporal dynamics of the 4-state model. a** Plots of the multiplication of transition weights and transition inputs. This illustrates the temporal patterns of the weighted transition regressors in the same three sessions of one mouse as in Fig. 4a (Background color marks block bias: pink = right-biased, blue = left-biased). **b** The histogram illustrates the distribution of inferred state changes per session across sessions exceeding a duration of *T*. **c** The anticipated duration of stays, or dwell time histograms, in different states for all mice: this was achieved by utilizing the derived transition matrix for each individual mouse. Source data are provided as a Source Data file.

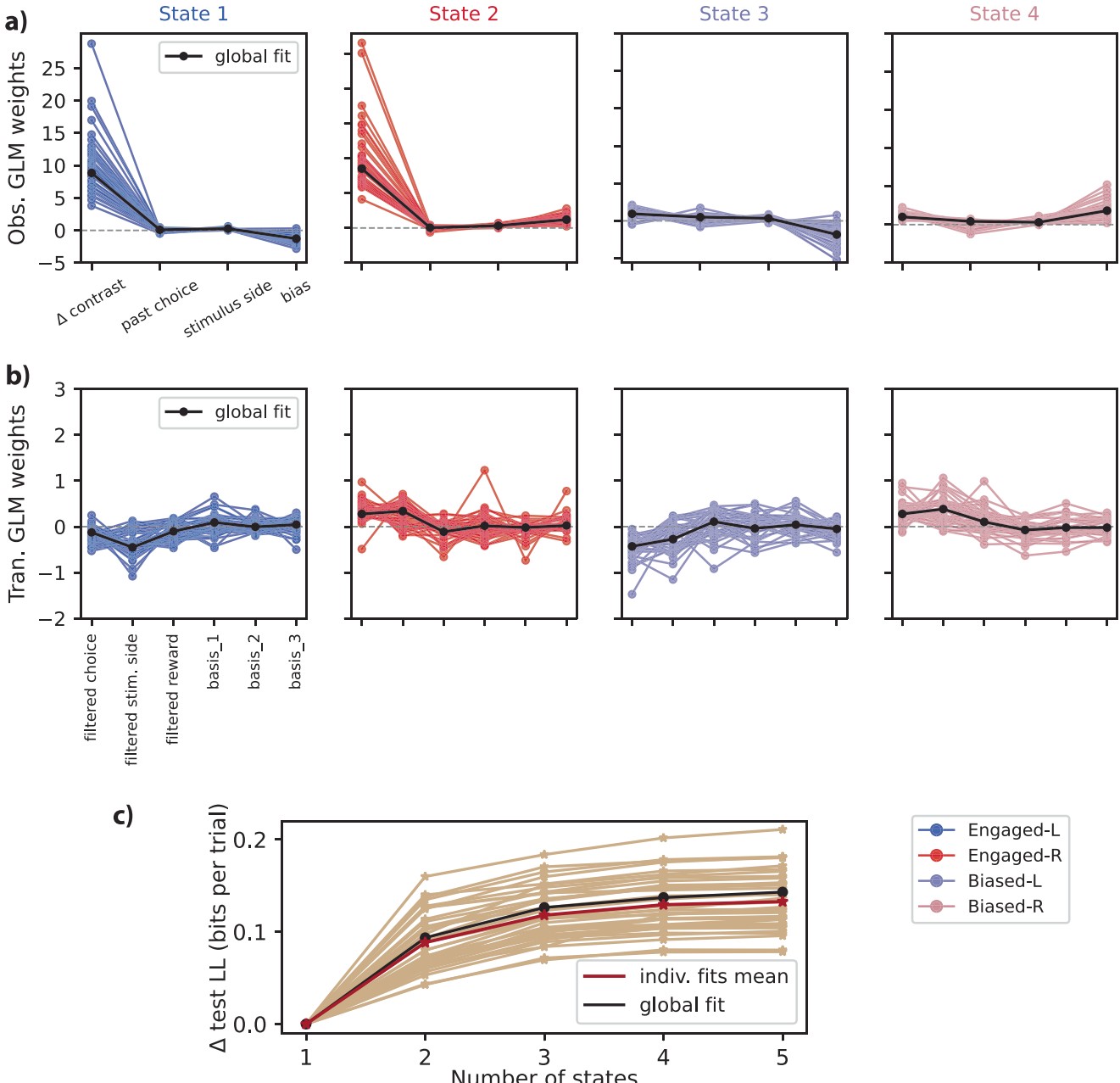

**Fig. 6 | Analysis of data from each individual mouse in the IBL dataset.**
**a** Observation weights for the Bernoulli GLM corresponding to distinct states within the four-state model across all IBL animals, highlighting the similarity and trends in state-specific observation parameters. **b** Transition weights for the Multinomial GLM of the same model across all animals, showcasing the dynamics and consistency of transitioning between different states for all animals. **c** Test log-likelihood (LL) variation for each mouse in the population, illustrated against the number of states. Each trace represents an individual mouse, with the solid red line indicating the average across all individual fits and the black solid line showing the test log-likelihood for the global fit. **a**–**c** Unit of study: mouse. $n = 37$ mice (biological replicates; independent animals from the IBL dataset). Each point/curve corresponds to one mouse. Panel c shows held-out test log-likelihood per mouse across model sizes (fivefold cross-validation; folds at the session level). Source data are provided as a Source Data file. Stim. stimulus, Obs. observation, Tran. transition, indiv. individual.

positive correlation between filtered reward and the probability of disengagement states.

To assess the effectiveness of GLM-T in capturing trial-dependent dynamics, we compared model performance using synthetic data. Figure 7d presents the test log-likelihood comparison for models fitted to data sampled from a generative model without GLM-T (left) and a model with GLM-T (right). When trial-dependent transitions are present in the data, the GLM-T model performs better, improving by 0.009 bits per trial. Conversely, when transitions are independent of trials, the non-GLM-T performs almost similarly to the model with GLM-T. This net advantage becomes particularly significant in large datasets, making the GLM-T model more suitable for capturing structured behavioral dynamics. This improvement of 0.009 bits per trial when the data has trial-dependent transitions means that for a dataset with 5000 trials, the data would be $-3.52 \times 10^{13}$ times more probable under the GLM-T model than under the non-GLM-T model. Figure 7e compares the inferred states from both models, with most data points lying above the $y = x$ reference line (red dashed), indicating that the GLM-T model assigns higher probabilities to states compared to the non-GLM-T model, demonstrating its confidence in detecting states and its improved performance.

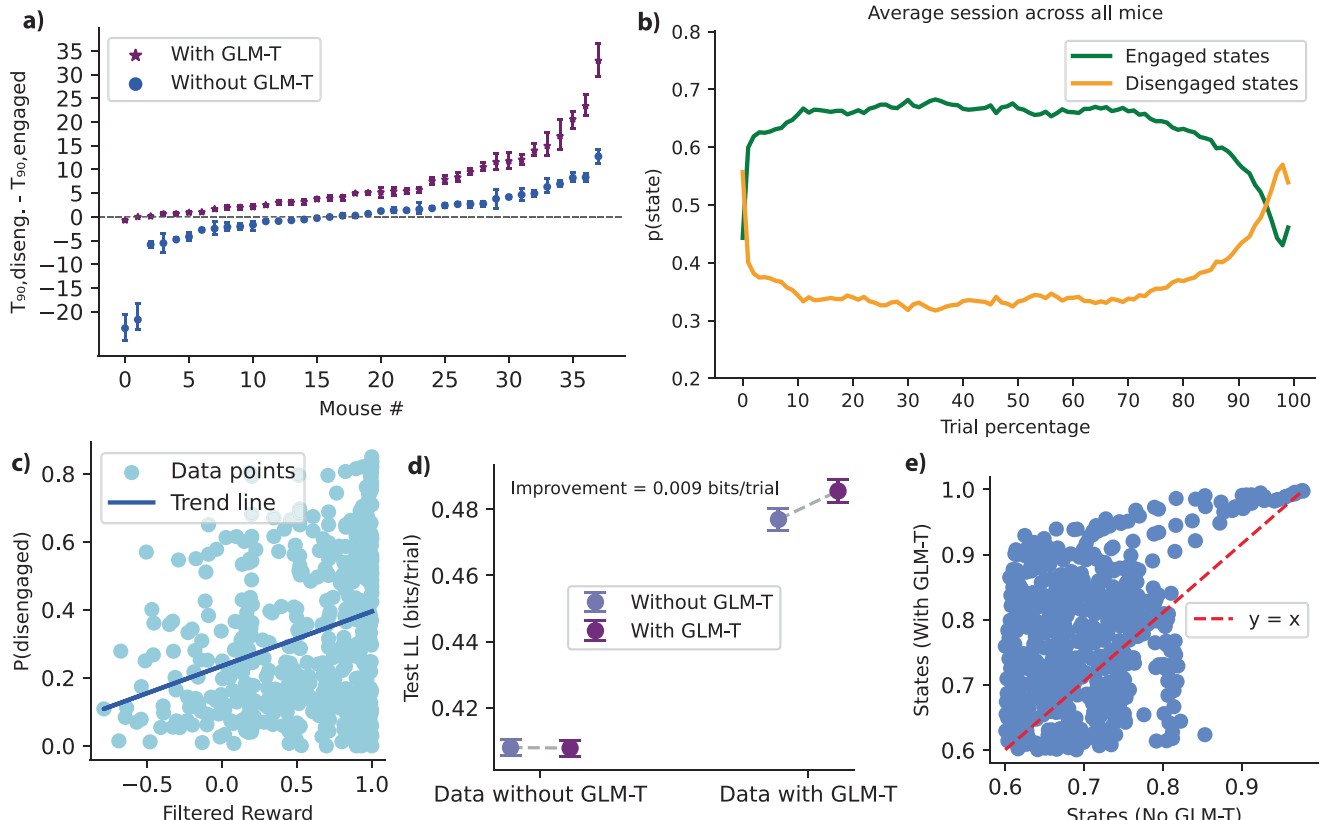

**Fig. 7 | Model performance. a** Difference in 90th percentile response time (disengaged - engaged) across mice for models with (purple) and without GLM-T (blue), indicating a higher separation of engaged and disengaged states in terms of response times for the model with GLM-T. This was for $n = 37$ mice (biological replicates; independent IBL animals). Sessions/trials are replicates aggregated within mouse. Points show the per-mouse difference in the 90th percentile response time. Error bars denote 95% bootstrap percentile confidence intervals (2.5–97.5th) computed from 5000 resamples of trials within each mouse. **b** Average state probabilities across trials for all mice, showing an initial warm-up effect where mice start in disengaged states (orange) before transitioning to engaged states (green). Additionally, this suggests a link between disengagement and satiety, as greater past rewards predict transitions to disengaged states, leading to an increased occupancy of disengagement states toward the end of the session when mice have accumulated more rewards throughout the experiment. **c** Relationship between filtered reward and the probability of being in the disengaged state, showing a positive correlation between them (light blue points are data points and blue line is the trend line). **d** Test log-likelihood (LL) comparison between models for synthetic data sampled from a model without GLM-T (left) and a model with GLM-T (right). When data has trial-dependent transitions, the GLM-T model performs better, improving by 0.009 bits per trial. Conversely, when transitions are independent of trials, the non-GLM-T model performs almost similar. The relative improvement, shows that it provides a meaningful advantage for data with transitions. Models were run on synthetic data produced for $n = 37$ mice by reconstructing each mouse's best-fitting model and simulating session-by-session using the true stimulus sequences. Data are presented as mean test LL; error bars denote ± standard deviation across 5-fold cross-validation. **e** Scatter plot of inferred states with and without GLM-T, with a reference $y = x$ line (red dashed). Most points lie above the red line, indicating that GLM-T states are more probable than non-GLM-T states. Source data are provided as a Source Data file. Diseng., Disengaged.

These results collectively demonstrate that incorporating GLM-T leads to better performance and more structured and behaviorally relevant state inferences, making it a more effective framework for modeling decision-making dynamics. Fig. S7–S10 further support this by showing that GLM-T improves the capture of structured state transitions, enhances the stability and clarity of posterior state probabilities, and enables dynamic, trial-dependent transition probabilities that align with block structures, whereas models without GLM-T fail to capture these behavioral dynamics effectively. Furthermore, GLM-O weights were similar across models with and without the GLM-T component (Fig. S7a). This shows that the GLM-T regulates transitions between behavioral states without substantially modifying the states themselves, as described by the GLM-O. Fig. S11 shows that transition probabilities into a given state are nearly identical across different preceding states, for IBL data. This confirms that our key conclusions are not sensitive to this aspect of the model.

## Discussion

In summary, this paper presents a GLM-HMM framework for analyzing mice's decision-making behavior in non-stationary environments. This model offers a nuanced understanding of transitional patterns of behavioral states and is structured as an HMM, featuring two sets of per-state GLMs: one for observations and one for transitions. This design provides flexibility in capturing the impact of independent sets of covariates on mouse choices and state transitions. When applied to the extensive IBL dataset, our analysis of 123 mice, including all sessions from 37 of them, highlights the superior performance and interpretability of the four-state GLM-HMM, revealing intricate patterns in the data. This study underscores the model's effectiveness in representing animal decision behavior and transition probabilities between states, especially in scenarios with variable stimulus probabilities.

We observed that the model with an additional GLM for transitions performs better than a basic model without a transition GLM, as evidenced by the test log-likelihood plot (Fig. 2a). For the observation model, stimulus and mice bias played significant roles, identifying two engaged states with low biases relative to the stimulus and two disengaged states with pronounced biases. For the transition model, we selected filtered choice, stimulus side, and reward as key covariates based on their theoretical relevance, and they effectively captured transitions between left- and right-bias states as well as between

engaged and disengaged states. Our analyses revealed that mice preferentially used left-bias strategies during left-bias stimulus blocks and right-bias strategies during right-bias stimulus blocks, achieving high performance even in disengaged states by biasing choices toward the side with greater prior probability.

While our work has primarily focused on the 4-state model, our findings are not limited to this configuration. Results and insights can be generalized to a 5-state model, detailed in the Supp., where the fifth state shows the effect of past choice in modeling decision behavior. Although the cross-validated log-likelihood gain is slightly higher for the 5-state model, the overall results remain similar in several aspects. Furthermore, our results indicate that past rewards play a key role in driving transitions to disengaged states, supporting the interpretation that satiety contributes to disengagement. The four-state model effectively captures this pattern, and the five-state model provides additional flexibility, suggesting that disengagement may encompass nuanced behavioral variations (Fig. S2b). While satiety is a likely contributor, disengagement may also reflect adaptive shifts in strategy or transient reductions in task engagement. Rather than indicating uncertainty, these findings highlight the richness of behavioral states and suggest that disengagement is not a monolithic process but may be shaped by multiple internal and external factors. Future work could further refine this interpretation by integrating physiological and neural measures to better characterize the mechanisms underlying disengaged states.

Looking ahead, it would be useful to investigate potential groups or clusters in the individual fits for different animals. For example, are some mice consistently less engaged than others, and do these engagement rates persist across sessions? Additionally, it would be valuable to explore a comparative analysis between the discrete state model presented in this paper (GLM-HMM) and a model incorporating continuously changing states over time. This comparison could offer deeper insights into decision-making behavior and the underlying neural processes[45–50]. On the other hand, it would be useful to explore other covariates that may influence strategy switching but were not included in our current model. While we selected covariates based on prior literature and theoretical reasoning, a more data-driven approach could help identify additional predictors, such as response time, movement vigor, or task engagement metrics, to capture decision-making dynamics in more detail.

Future work includes advancing our understanding of neural activity[51,52] and neural decoding[53–56] in relation to the discrete behavioral states identified in our GLM-HMM framework. Unraveling the intricate patterns of neural and behavioral correlations promises to be a pivotal direction in neuroscience research[57–62], especially in the context of decision-making processes. Analyzing and comparing neural modes across different tasks while considering a wide-ranging behavioral repertoire[63] can provide valuable insights into the animals decision-making strategies. Using state space methods for neural data analysis[64] and comparing these neural states with behavioral data could further enhance our understanding of how different factors influence decision-making.

Additionally, future work could involve investigating how different brain regions communicate using both neural and behavioral data, analyzed through GLM-HMM or other modeling approaches. Incorporating data from various brain areas[65,66] could enhance our understanding of animal decision-making strategies. Extending this research to map multi-region and brainwide spontaneous activity to animal behavioral patterns[6] and brain states could provide deeper insights into the mechanisms driving these behaviors.

Another promising direction could be focusing on modeling real-time neural activity and its correlation to online behavioral data and states, which could enhance the performance of brain-machine interfaces. While there has been some work on analyzing animal data in real-time[67–72], particularly for brain-machine interfaces[73,74] and their application to control brain states[75], there has been limited effort on modeling how this neural or behavioral data is encoded or decoded in real-time.

On the other hand, hierarchical models for analyzing animal data[76] offer a powerful means to capture the complexity of behavioral data by incorporating multiple levels of variability. These models can account for individual differences in behavior, learning, and strategy adoption while integrating neurophysiological data, enabling a deeper understanding of decision-making. Exploring the hierarchical version of GLM-HMM shows promise for uncovering the complex links between neural activity, behavioral states, and animal strategies.

In essence, our GLM-HMM framework provides a robust tool for understanding animal decision-making, highlighting the importance of considering both observation and transition dynamics. This approach uncovers links between mice's strategies, input regressors, and state transitions, emphasizing the complexity of adaptive behavior. It lays a foundation for future research in dynamic decision-making models, offering valuable insights into the mechanisms and data patterns underlying animal behavior.

## Methods
### Hidden Markov Model
An HMM is a statistical model for time series data that is governed by hidden or latent factors that cannot be directly observed. The events that we observe are called observations, and the underlying, unobservable factors driving them are referred to as latent states[29,30,77].

Therefore, HMM consists of two stochastic components: one governing the latent states, and another governing the observations. The latent process component satisfies the Markovian property. A transition probability matrix $A \in \mathbb{R}^{K \times K}$, where, at trial $t$, the element corresponding to state $j$ and state $k$ presents the transition probability between those two states and can be written as:

$$P(z_{t+1} = k | z_t = j) \tag{7}$$

and an observable state-dependent component $\{y_t\}_{t=1}^{T}$, for which we have:

$$P(y_t | y_1, y_2, \ldots, y_{t-1}, z_t) = P(y_t | z_t) \tag{8}$$

where $T$ is the number of considered trials. Generally, in HMM, the probability distribution of the observed symbols is based on the underlying, unobserved states of the system, following the principles of a Markov chain. In many cases, observations can be grouped into different classes, which can provide more insightful information than the individual observations themselves. In such situations, it becomes advantageous to model these observations using both the observable and unobservable aspects of HMM.

### Bernoulli GLM
The Bernoulli Generalized Linear Model (GLM) is a statistical model designed for binary data, where the response variable can take values of 0 or 1. It belongs to the broader GLM family, which includes models like Poisson, Gaussian, and Dirichlet. The primary purpose of using Bernoulli GLMs here is to model the relationship between a mouse's expected decision and the relevant regressors for each trial.

In the Bernoulli GLM, the response variable follows a Bernoulli distribution, which is a discrete probability distribution. It takes the value 1 with probability $p$ (representing success) and 0 with probability $1-p$ (indicating failure). Estimating the probability $p$ in the Bernoulli GLM involves predictor variables, and a link function connects the mean of the Bernoulli distribution to the linear predictor.

The linear predictor is formed as a linear combination of the predictor variables and their respective coefficients. Subsequently, the link function is used to transform the linear predictor to the

probability scale. The logit link function stands as the most widely adopted link function for a Bernoulli GLM, and it can be expressed as $\log(p/(1-p)) = F\beta$, where $F$ corresponds to a predictor variable matrix, and $\beta$ represents a vector of coefficients.

In this study, we employed a Bernoulli GLM to analyze the animal's strategies, presented in Fig. 1e, in relation to various experiment regressors. It can map the binary values of the animal's decision to the weighted representations of the considered covariates. These weights serve to depict the inputs of the model in relation to the output, which is the animal's choice on each trial. Consequently, we can describe an observational GLM using the following equation, where the animal choice, denoted by $y$, can take a value of 1 or 0, indicating the mouse turning the wheel to the right or left-side, respectively:

$$p(y|z_t = k, x) \equiv \frac{\exp(y\mathbf{w}_k\mathbf{x})}{1 + \exp(\mathbf{w}_k\mathbf{x})}. \tag{9}$$

In this equation, as indicated by the notations, the presented GLM is solely associated with the observation covariates, $\mathbf{x}$, and observation weights $\mathbf{w}$ for state $k$.

The fitting procedure of the model involves utilizing a penalized maximum likelihood estimation. This estimation minimizes the sum of the priors on the transition and observation weights, in addition to a negative log-likelihood function, often referred to as the log-posterior. The prior corresponds to a normal distribution over the weights with a mean of zero and a variance of $\sigma^2$. The negative log of this prior can be represented as $\frac{1}{2}(\mathbf{w})^T\mathbf{w}$. The purpose of this prior is to impose a penalty on the model weights, thereby regularizing the model by discouraging excessively large weight values for the regressors[78]. Consequently, a relevant loss function can be defined as:

$$\text{Loss}(\mathbf{w}) \equiv -\log p(\mathbf{Y}|\mathbf{w}, \mathbf{X}) + \lambda\frac{1}{2}\mathbf{w}^T\mathbf{w}. \tag{10}$$

Here, $\log p(\mathbf{Y}|\mathbf{w}, \mathbf{X})$ represents the conditional probability of the output, which corresponds to the decisions made by the animals, given the model regressors. Also, the symbol $\lambda$ assumes the role of a hyperparameter that governs the regularization term's influence on the model. The log-likelihood function can be mathematically defined as follows:

$$\log p(\mathbf{Y}|\mathbf{w}, \mathbf{X}) = \sum_{t=1}^{T} \log p(y_t|\mathbf{w}, \mathbf{x}_t)$$
$$= \sum_{t=1}^{T} \left( y_t\mathbf{w}^\top\mathbf{x}_t - \log(1 + e^{\mathbf{w}^\top\mathbf{x}}) \right). \tag{11}$$

In this context, $\mathbf{Y}$ represents the observations from trial 1 to $T$, while $\mathbf{X}$ corresponds to the regressors for the GLM applied to the same trials.

The GLM can be fitted using the maximum likelihood or MAP estimation, and the resulting coefficients can subsequently be used to make predictions on new data. To assess the performance of the fitted model, we use cross-validation on held-out test data.

## Multinomial GLM

Another form of the GLM is the multinomial GLM, which we will employ to model how external covariates influence transitions between different states. This multinomial logistic regression, also known as softmax regression or maximum entropy classifier, serves as an extension of logistic regression to handle data with multiple categories. Multinomial GLMs are GLMs possessing the capability to analyze data from more than one category simultaneously. By modeling the relationship between independent variables and categorical dependent variables, this approach allows for the determination of the

likelihood associated with each category. In the field of neuroscience, multinomial GLMs are frequently employed to analyze the link between brain activity and behavior, facilitating the investigation of neural processes underlying different types of behavior.

The primary difference between Bernoulli and multinomial GLM lies in the number of categories the models handle. Bernoulli GLM is used for binary outcomes, modeling the probability of success (1) or failure (0), commonly employed in binary classification. Multinomial GLM, on the other hand, deals with data featuring more than two categories, modeling the probabilities of each category using a multinomial logistic function. The choice between them depends on whether you're working with binary or multiclass categorical data. In multinomial GLM, the expression for the conditional probability of observing a particular outcome $y$, given input variables $\mathbf{x}$ and a set of model parameters $\mathbf{w}$, is typically formulated as follows:

$$p(y = c|\mathbf{x}, \mathbf{w}) = \frac{\exp(\mathbf{w}_c^\top\mathbf{x})}{\sum_{j=1}^{C} \exp(\mathbf{w}_j^\top\mathbf{x})} \tag{12}$$

where $\mathbf{w}_j$ corresponds to the parameters associated with j-th outcome and $C$ is the number of possible outcomes in which for multinomial GLM $C$ is more than 2. The multinomial equation finds frequent application within the domain of multinomial logistic regression, serving as a means to model the probability distribution across numerous discrete outcomes or categories. This mathematical framework is foundational in a multitude of machine learning and statistical contexts, including but not limited to tasks such as text classification, image recognition, and addressing multiclass classification challenges.

In this paper, we delve into the exploration of the GLM-HMM, GLM-HMM, with multinomial GLM outputs, a method capable of estimating the likelihood of the next state (presented in Fig. 1e). Notably, the GLM-HMM framework offers the advantage of enabling each state to possess a multinomial GLM that effectively captures the intricate relationship between transition covariates, such as previous choice and previous reward, and the associated transition probabilities.

Therefore, we consider the animal's behavior at trial $t$ and proceed to calculate transition probabilities to various states at trial $t + 1$ utilizing a multinomial GLM. The model is adeptly structured to establish associations between the vector of model transition inputs and the unnormalized log probability of each potential future state. So our model presents transition probabilities using a multinomial GLM, and emission probabilities using a Bernoulli GLM, each defined by its own set of parameters.

## Structure of the GLM-HMM

GLM-HMM, or Generalized Linear Model-HMM, is a sophisticated probabilistic model in which the core structure combines the principles of HMM and GLM[31–38]. At its essence, GLM-HMM is defined by a dual-layered structure. The first layer comprises an HMM, a stochastic process with hidden states that transition over time. These hidden states capture latent information about the underlying dynamics of a system. The second layer incorporates GLMs, which govern the mapping from inputs to outputs, and this mapping is influenced by the current hidden state of the HMM. This dual-layer structure enables GLM-HMM to effectively model complex, sequential data where the relationship between observed outputs and input variables varies depending on the underlying, unobservable state. This capacity to incorporate state-dependent relationships within a probabilistic framework makes GLM-HMM an appropriate tool for analyzing and understanding temporal data in various domains.

In this manuscript, we present a GLM-HMM framework that incorporates both GLM observation and GLM transition models with independent covariates. This approach enables us to capture the temporal patterns of animal transitions between states, enhancing our

understanding of decision-making dynamics in mice within non-stationary environments. Specifically, we use a Bernoulli GLM for observations and a multinomial GLM for transitions. The Bernoulli GLM captures the roles of different regressors in animal choices, establishing the relationships between observed and predictor factors by defining the conditional distribution of the observed parameter based on its predictors. The multinomial component captures the intricate patterns associated with state transitions, characterizing the transition probabilities between the hidden states. Instead of a fixed transition matrix between states, our model uses a set of vectors, one for each state, to capture transitions into that state, with the averaged transition probability matrix presented in Fig. 3e. At each trial or time point in a dynamic environment, these probabilities govern the likelihood of transitioning from one state to another.

Therefore, in the context of a multinomial GLM, for the transition model, the GLM output reflects different probabilities for transitions between states. In this case, $\mathbf{x}_t^{\mathrm{tr}}$ presents the transition covariates at trial $t$ and $\mathbf{w}_k^{\mathrm{tr}}$ is the transition weights associated with state $k$. So for the transition probability to state $k$ at trial $t$, denoted as $z_t$, we can write:

$$p(z_t = k | \mathbf{x}_t^{\mathrm{tr}}) = \frac{\exp\{(\mathbf{w}_k^{\mathrm{tr}})^\top \mathbf{x}_t^{\mathrm{tr}}\}}{\sum_{j=1}^{K} \exp\{(\mathbf{w}_j^{\mathrm{tr}})^\top \mathbf{x}_t^{\mathrm{tr}}\}} \qquad (13)$$

here, $K$ represents the total number of states.

Furthermore, the probability of transitioning from state $i$ to state $j$ at time $t$, given transition covariates $\mathbf{x}_t^{\mathrm{tr}}$, is modeled as:

$$p(z_t = j | z_{t-1} = i, \mathbf{x}_t^{\mathrm{tr}}) \propto \exp\left(B_{ij} + \mathbf{w}_j^{\mathrm{tr}\top} \mathbf{x}_t^{\mathrm{tr}}\right) \qquad (14)$$

where $B_{ij}$ is a learned baseline transition logit from state $i$ to $j$. Also, $\mathbf{w}_j^{\mathrm{tr}}$ is a vector of covariate weights that modulate all incoming transitions to state $j$.

The log-likelihood function in a multinomial GLM is rooted in the multinomial distribution. It quantifies the probability of observing categorical outcomes based on the predictor variables and model parameters. This log-likelihood function is conventionally formulated as follows:

$$\begin{aligned}
\log L(\mathbf{w}^{\mathrm{tr}}) &= \sum_{i=1}^{N} \log\left(\frac{\exp\{(\mathbf{w}_{k_i}^{\mathrm{tr}})^\top \mathbf{x}_i^{\mathrm{tr}}\}}{\sum_{j=1}^{K} \exp\{(\mathbf{w}_j^{\mathrm{tr}})^\top \mathbf{x}_i^{\mathrm{tr}}\}}\right) \\
&= \sum_{i=1}^{N} \left((\mathbf{w}_{k_i}^{\mathrm{tr}})^\top \mathbf{x}_i^{\mathrm{tr}} - \log \sum_{j=1}^{K} \exp\{(\mathbf{w}_j^{\mathrm{tr}})^\top \mathbf{x}_i^{\mathrm{tr}}\}\right)
\end{aligned} \qquad (15)$$

in which $N$ is the total number of outputs and $\mathbf{x}_i^{\mathrm{tr}}$ represents the transition regressors for the i-th output. The other details of the mathematical description of this model will be explained in the upcoming sections. This comprehensive framework provides a robust tool for analyzing flexible decision-making and the underlying mechanisms in non-stationary environments.

## Model inputs

In this framework, we incorporated several covariates into the model, some of which are shown in Figs. 1d and 3a, and will be elaborated upon in detail here.

Observation model covariates: For the GLM observation, the covariates were stimulus, previous choice, previous stimulus and bias. The stimulus, in the experiment, was defined as the contrast of a sinusoidal grating, varying between 0% and 100% brightness. To further normalize the stimulus, it was divided by the standard deviation of the trials across sessions. Also, the past choice was defined as a binary variable with values of −1 or 1, depending on whether the mouse's previous choice was left or right, respectively. On the other hand, the previous stimulus was defined as the behavior of the mouse when rewarded. If the mouse received a reward, it continued to make the

same decision; however, when no reward was given, it changed its behavior. The previous stimulus values were binary, denoted as -1, +1. Finally, bias is a covariate that captures the animal's inherent tendency to choose left or right, representing an internal preference or offset that exists independently of external factors such as the stimulus or choice history, and its value was set to 1 for all trials.

Transition model covariates: The covariates for the GLM transition included a filtered version of the previous choice, previous stimulus, previous reward, and three basis coefficients. Throughout this paper, the term filtered covariate denotes a covariate filtered with an exponential filter, enabling the consideration of a temporally filtered regressor to consider the impact of the regressors in the transition between states. Therefore, the filtered previous choice and previous stimulus are temporally filtered versions of similar GLM observation covariates. The previous reward value was set to -1 or 1, based on whether the previous animal's decision was correct or not, respectively. Furthermore, three bases were defined as linearly independent vectors to represent the initial 100 trials of each session. These covariates aimed to capture the animal's warm-up effect within the model.

## Setting a prior on the model parameters

We use MAP estimation to fit model parameters, denoted as $\Theta = \{\mathbf{w}^{\mathrm{tr}}, \mathbf{w}^{\mathrm{ob}}, \boldsymbol{\pi}\}$. These parameters consist of the initial state distribution, transition weights, and observation weights for all states. The observation weights are shown in Fig. 2b, and the transition weights are presented in Fig. 3b, c.

To implement our approach, we employed the EM method, as introduced by Dempster et al.[79]. This iterative technique enables the determination of parameters within a given model by maximizing the likelihood of the data, given the specific parameters. Previous works[31,40,41] have successfully applied the EM approach to HMMs integrated with external regressors. The parameter estimation using the EM method involves an iterative process with two essential steps: the expectation step, where parameter expectations are computed, and the maximization step, optimizing data likelihood based on those parameters. This iteration continues until optimal parameter values are determined.

In this study, we incorporated a Dirichlet prior to model the initial state distribution $\boldsymbol{\pi}$. As for the GLM, we employed independent zero-mean Gaussian priors for the observation and transition weight vectors, with variances denoted as $\sigma_{\mathrm{ob}}^2$ and $\sigma_{\mathrm{tr}}^2$, respectively. Larger values of these variances signify a flatter prior distribution. The entire set of model choices and inputs is expressed as $\mathcal{F} \equiv \{\mathbf{Y}, \mathbf{X}^{\mathrm{ob}}, \mathbf{X}^{\mathrm{tr}}\}$. Here, $\mathbf{X}^{\mathrm{ob}} = \mathbf{x}_1^{\mathrm{ob}}, \ldots, \mathbf{x}_T^{\mathrm{ob}}$ represents the observation input vectors, and $\mathbf{X}^{\mathrm{tr}} = \mathbf{x}_1^{\mathrm{tr}}, \ldots, \mathbf{x}_T^{\mathrm{tr}}$ corresponds to the transition input vectors. The animal choices for the specified trials are denoted as $\mathbf{Y} = y_1, \ldots, y_T$. The prior distribution considered for our GLM-HMM, $p(\Theta)$, is as follows:

$$p(\Theta) \equiv p(\boldsymbol{\pi}) p(\mathbf{w}_k^{\mathrm{ob}}) p(\mathbf{w}_k^{\mathrm{tr}}). \qquad (16)$$

In this context, the model parameters encompass three distinct categories, namely, the initial state distribution, the state-specific observation weights and state-specific transition weights denoted by $\mathbf{w}_k^{\mathrm{ob}}$ and $\mathbf{w}_k^{\mathrm{tr}}$, respectively. Here, $k$ denotes the number of states, ranging from 1 to $K$. The initial state distribution is represented by $\boldsymbol{\pi} \in \mathbb{R}^K$. The prior distributions for the observation and transition weights in the GLM are both assumed to follow normal distributions, as expressed in the following equation:

$$p(\boldsymbol{\pi}) = \mathrm{Dirichlet}(\boldsymbol{\pi}|\gamma_\pi), \quad p(\mathbf{w}_k^{\mathrm{ob}}) = \prod_{j=1}^{K} \mathcal{N}(\mathbf{w}_j^{\mathrm{ob}}|0, \sigma_{\mathrm{ob}}^2 I), \quad p(\mathbf{w}_k^{\mathrm{tr}})$$

$$= \prod_{j=1}^{K} \mathcal{N}(\mathbf{w}_j^{\mathrm{tr}}|0, \sigma_{\mathrm{tr}}^2 I). \qquad (17)$$

The prior value for the initial state distribution was assigned as $\gamma_\pi = 1$. To determine the optimal hyperparameter value for the model prior on weights, we conducted a grid search over a range of values, specifically $\sigma \in \{0.25, 0.5, 1, 2, 4, 8, 16\}$. We employed a held-out validation set to compare the NLL values for the different $\sigma$ values within the specified set. Subsequently, the value of $\sigma$ yielding the highest NLL on the IBL data was selected, and it was found to be $\sigma = 4$.

The GLM-HMM employs an HMM, encompassing distinct sets of weights for both transition and observation aspects. Within the model, each state is governed by a state-specific Bernoulli GLM and a multinomial GLM, representing the animal's decision-making behavior concerning choice probability and the probability of transitioning between states, respectively. In this context, we used the notations GLM-O and GLM-T to represent GLM observation and GLM transition, respectively.

The transition multinomial GLM involves weights that associate relevant regressors, denoted as $\mathbf{x}^{tr}$, with the probabilities of transitioning between states. These transition probabilities are not fixed values and are contingent upon the combination of related regressors and the current state. They are represented by a matrix $A \in \mathbb{R}^{K \times K}$, where, at trial $t$, the element corresponding to state $j$ and state $k$ presents the transition probability between those two states and can be written as:

$$\alpha_{jk} = P(z_{t+1} = k | z_t = j, \mathbf{x}_t^{tr}). \tag{18}$$

It is pertinent to acknowledge that the observation Bernoulli GLM plays a crucial role in determining the observation weights and characterizing the decision behavior as a function of the input observation regressors.

Ultimately, in the context of mice decisions, along with transition and observation covariates, the primary goal of the EM algorithm is to optimize the log posterior of the model parameters. The log-posterior can be mathematically expressed as follows:

$$\log P(\Theta | \mathcal{F}) = \log P(\mathbf{Y} | \mathbf{X}^{ob}, \mathbf{X}^{tr}, \Theta) + \log P(\Theta) + const.. \tag{19}$$

## Model fitting

The GLM-HMM employs the EM algorithm[40] to optimize model parameters for maximizing the likelihood of observed data[78]. This iterative algorithm consists of two main steps: the E-step, where the expected complete-data log-likelihood is computed based on parameter estimates and observed data, and the M-step, which maximizes model parameters based on these expectations. The likelihood is calculated using the forward-backward approach[78], a dynamic programming algorithm. The EM algorithm continues iteratively until convergence is achieved, ensuring an accurate parameter estimation process.

To elaborate further, during each trial and based on the specified GLM-HMM parameters, we compute the joint probability distribution encompassing both the states and the animals' decisions (left or right). Subsequently, the log-likelihood of the model is evaluated using this joint probability distribution. This relationship can be expressed in the following manner:

$$\log \left[ p(\mathbf{Y} | \theta, \mathbf{X}^{ob}, \mathbf{X}^{tr}) \right] = \log \left[ \sum_z p(\mathbf{Y}, \mathbf{Z} | \theta, \mathbf{X}^{ob}, \mathbf{X}^{tr}) \right] \tag{20}$$

in which as mentioned, $\mathbf{X}^{ob} = \mathbf{x}_1^{ob}, ..., \mathbf{x}_T^{ob}$ represents the observation covariates, and $\mathbf{X}^{tr} = \mathbf{x}_1^{tr}, ..., \mathbf{x}_T^{tr}$ represents the transition covariates. Additionally, we have a set of latent states denoted as $\mathbf{Z} = z_1, ..., z_T$, and corresponding observations for these states denoted as $\mathbf{Y} = y_1, ..., y_T$.

In the model, $\mathbf{X}^{ob}$ and $\mathbf{X}^{tr}$ capture relevant information related to the observations and transitions, respectively. These covariates play a crucial role in characterizing the underlying dynamics of the system

under consideration. The latent states, $\mathbf{Z}$, represent unobservable or hidden variables that drive the observed data. They are essential components of the model, as they provide insights into the underlying processes governing the observed phenomena. The observations, $\mathbf{Y}$, are the data collected from the system, corresponding to each specific latent state in $\mathbf{Z}$. These observed data points are used in the model to estimate and infer the hidden states and the model parameters.

In the GLM-HMM, state-dependent GLM-T weights represent transition regressor weights, and GLM-O weights represent the significance of observation covariates. These weights' patterns differ across distinct states. In the Bayesian context, where prior information exists for unknown parameters, the EM algorithm can be used to compute the mode of the posterior probability distribution, facilitating parameter estimation.

In the context of probabilistic graphical models and variational inference, the E-step is often associated with variational lower bounds. Specifically, when dealing with intractable posterior distributions or complex models, the E-step aims to maximize a lower bound on the log-likelihood, rather than the log-likelihood itself. This lower bound is often referred to as the Evidence Lower Bound or the variational lower bound. Here, during the E-step of the EM algorithm, a lower bound on the right-hand side of the objective function referred to as Eq. (20)[79,80] is maximized. Subsequently, in the M-step, the expected log-likelihood obtained from the E-step is maximized with respect to the GLM-HMM parameters. In the following sections, we will detail the calculations for both the E-step and M-step of this estimation approach. These steps play a crucial role in iteratively refining the parameter estimates until convergence is achieved, enabling the determination of the mode of the posterior probability in the Bayesian setting.

By defining the Bernoulli GLM distribution as $p(y_t | z_t = k, \mathbf{x}_t^{ob}, \mathbf{w}_k^{ob})$, we obtain the following equation for the expected log-likelihood during the E-step:

$$\sum_\mathbf{z} p\left(\mathbf{z} | \mathcal{F}, \Theta^{(t-1)}\right) \log p\left(\mathbf{Y}, \mathbf{z} | \Theta, \mathbf{X}^{ob}, \mathbf{X}^{tr}\right). \tag{21}$$

In this equation, we can express the model joint distribution, $p\left(\mathbf{Y}, \mathbf{z} | \Theta, \mathbf{X}^{ob}, \mathbf{X}^{tr}\right)$, as:

$$p(z_1) p(y_1 | z_1, \mathbf{x}_1^{ob}) \prod_{t=2}^{T} p(z_t | z_{t-1}, \mathbf{x}_t^{tr}) p(y_t | z_t, \mathbf{x}_t^{ob}). \tag{22}$$

To calculate this expected log-likelihood, the E-step uses a forward-backward approach[81] to estimate the single and joint posterior state probabilities. We are going to explain each one separately here. So the process involves calculating the single posterior state probability at trial $t$, given by:

$$p\left(z_t = k | \mathbf{Y}, \mathbf{X}^{ob}, \mathbf{X}^{tr}, \Theta^{(t-1)}\right). \tag{23}$$

We show this probability by $\phi_{t,k}$. In the forward-backward approach, the E-step iteratively calculates the posterior probability of the mice decisions by going through a loop. As a result, the single posterior state probability can be decomposed into the following equation:

$$\phi_{t,k} \equiv \frac{p\left(\mathbf{Y}_{[0:t]}, z_t = k | \{\mathbf{X}^{ob}, \mathbf{X}^{tr}\}_{[1:t]}, \Theta^{(t-1)}\right) p\left(\mathbf{Y}_{[t+1:T]} | z_t = k, \mathbf{X}_{[t+1:T]}^{ob}, \Theta^{(t-1)}\right)}{p\left(\mathbf{Y} | \mathbf{X}^{ob}, \Theta^{(t-1)}\right)}$$
$$\tag{24}$$

in which $\Theta^{(t-1)}$ is the parameters of the model at trial $t-1$.

In the E-step of the estimation process, the forward-backward algorithm is used to compute the expectation of the desired function. This two-stage message-passing algorithm is also known as the filtering

process during the forward pass[40]. The forward-backward algorithm, through its filtering process, computes the posterior probabilities for each time point, enabling the estimation of the latent states' influence on the observed data. In the following, we will explain the forward and backward passes of the algorithm.

Forward pass: In the E-step, the forward-backward algorithm is employed to obtain the expectation of the desired function. This is a two-stage message-passing algorithm[40] and the forward pass is called filtering. By assuming we have GLM-HMM parameters and observation data, to calculate the E-step, we should form the posterior probability as a function of latent states.

The objective of the E-step is to infer the hidden state sequence, given the available observations and the current parameter estimates. In the above equation for the first trial, the posterior probability is obtained by $\pi_k p(y_1|z_1 = k, \mathbf{x}_1^{ob}, \mathbf{w}_k^{ob})$.

Here, $p(y_1|z_1 = k, \mathbf{x}_1^{ob}, \mathbf{w}_k^{ob})$ is the Bernoulli GLM distribution for observations (GLM-O). In this equation, if we consider the posterior probability of the mice decisions, $\alpha_{t,k}$, from trial 1 to $t$ as:

$$\alpha_{t,k} = p\left(\mathbf{Y}_{[1:t]}, z_t | \{\mathbf{X}^{ob}, \mathbf{X}^{tr}\}_{[1:t]}\right). \tag{25}$$

Based on this definition, for the first trial, we can call the posterior probability as $\alpha_{1,k}$. Then using earlier calculations, for upcoming trials until $T$, we can compute the posterior probabilities, $\alpha_{t,k}$, as:

$$\sum_{j=1}^{K} \alpha_{t-1,j} \alpha_{jk} p\left(y_t | z_t = k, \mathbf{x}_t^{ob}, \mathbf{w}_k^{ob}\right). \tag{26}$$

Backward pass: In the forward-backward algorithm, the backward step complements the forward pass, utilizing previously obtained information to update latent state probabilities. Named backward due to its reverse chronological order, it calculates updates essential for the M-step. The backward pass relies on the joint posterior distribution of two latent states, contributing to refined GLM-HMM parameter estimates. Its primary goal is to determine the posterior probability of observed mouse data for all states and trials beyond the current trial. This probability is represented as $\beta_{t,k}$, where $t$ denotes the trial index and $k$ refers to a specific state. This can be written as:

$$p\left(\mathbf{Y}_{[t+1:T]} | z_t = k, \mathbf{X}_{[t+1:T]}^{ob}\right). \tag{27}$$

Therefore, for trial $T$, we have $\beta_{T,k} = 1$. Also, we can compute the posterior probability, $\beta_{t,j}$, for trials $\{T-1, \ldots, 1\}$ as:

$$\sum_{k=1}^{K} \beta_{t+1,k} \alpha_{jk} p\left(y_{t+1} | z_{t+1} = k, \mathbf{x}_{t+1}^{ob}, \mathbf{w}_k^{ob}\right). \tag{28}$$

Here, it should be considered that a Bernoulli GLM distribution for observations can be written as $p(y_{t+1}|z_{t+1} = k, \mathbf{x}_{t+1}^{ob}, \mathbf{w}_k^{ob})$. By incorporating the information from both the forward and backward passes, the model gains a better understanding of the system's underlying dynamics and is better equipped to estimate the latent states' influence on the observed data, leading to improved parameter estimation in the M-step of the EM algorithm. So, by employing the forward-backward approach, we have:

$$\phi_{t,k} = \frac{\alpha_{t,k} \beta_{t,k}}{\sum_{k=1}^{K} \alpha_{T,k}}. \tag{29}$$

On the other hand, for the joint posterior state probability $\mu_{t,j,k}$, we can write:

$$\frac{\alpha_{t,j} \alpha_{jk} \beta_{t+1,k} p\left(y_{t+1} | z_{t+1} = k, \mathbf{x}_{t+1}^{ob}, \mathbf{w}_k^{ob}\right)}{p\left(\mathbf{Y} | \mathbf{X}_{[1:T]}^{ob}, \Theta^{(t-1)}\right)} \tag{30}$$

and by substituting the denominator using the definition of posterior probabilities, we have:

$$\frac{\alpha_{t,j} \alpha_{jk} \beta_{t+1,k} p\left(y_{t+1} | z_{t+1} = k, \mathbf{x}_{t+1}^{ob}, \mathbf{w}_k^{ob}\right)}{\sum_{k=1}^{K} \alpha_{T,k}}. \tag{31}$$

Therefore, to calculate the expected log-likelihood, we should use the results of the posterior state probabilities which were obtained using the forward-backward algorithm. Therefore, for a given trial $t$ and state $k$, the posterior state probability is defined by $\phi_{t,k} \equiv p\left(z_t = k | \mathcal{F}, \Theta^{(t-1)}\right)$ and the joint posterior state distribution for states at trials $t$ and $t+1$ is given by $\mu_{t,j,k} \equiv p\left(z_{t+1} = k, z_t = j | \mathcal{F}, \Theta^{(t-1)}\right)$. In conclusion, by considering the single and joint posterior state probabilities, $\phi_{t,k}, \mu_{t,j,k}$, we can rewrite the Eq. (21) as:

$$\sum_{t=1}^{T} \sum_{j=1}^{K} \sum_{k=1}^{K} \mu_{t,j,k} \log \alpha_{jk} + \sum_{t=1}^{T} \sum_{k=1}^{K} \phi_{t,k} \log p(y_t | z_t = k, \mathbf{x}_t^{ob}, \mathbf{x}_t^{tr}, \mathbf{w}_k^{ob}, \mathbf{w}_k^{tr}) + \text{Init}. \tag{32}$$

Here, Init is the term related to the initialization and is equal to $\sum_{k=1}^{K} \phi_{1,k} \log \pi_k$.

M-step: The M-step of the Expectation-Maximization algorithm updates the GLM-HMM parameters by utilizing the posterior probabilities calculated during the E-step. The objective of the EM algorithm is to minimize the negative log-likelihood function, which is augmented with a prior on the GLM-HMM weights.

L-BFGS-B (Limited-memory Broyden-Fletcher-Goldfarb-Shanno with Bound constraints) is a popular optimization algorithm used to solve unconstrained and bound-constrained nonlinear optimization problems. It's an extension of the BFGS algorithm, which is a quasi-Newton method for unconstrained optimization. L-BFGS-B is particularly useful when dealing with optimization problems where the variables have certain bounds or constraints. These constraints can be upper and lower bounds on the variables, and L-BFGS-B is designed to handle such constraints. So L-BFGS-B is a second-order optimization method that approximates the Hessian matrix, leading to faster convergence and accurate parameter estimates. Here, the EM algorithm employs the L-BFGS-B approach, which belongs to the class of Quasi-Newton optimization methods.

During the M-step, the EM algorithm maximizes the expected log-likelihood, computed through the forward-backward algorithm during the E-step. This optimization process aims to identify the best parameters for the GLM-HMM. Consequently, in each iteration of the EM algorithm, the initial state distribution $\pi$ is updated to refine the model's representation of the latent state dynamics.

By iteratively performing the E-step and M-step, the EM algorithm iteratively refines the parameter estimates until convergence is achieved, enabling the model to accurately capture the underlying dynamics and relationships in the data, leading to improved parameter estimation and enhanced model performance. The posterior plots for a few sessions of a mouse's data are depicted in Fig. 4a. Also, for the same sessions, the plots of the multiplication of transition weights and transition inputs are shown in Fig. 5a, illustrating the temporal patterns of the weighted transition regressors.

As mentioned, the EM algorithm calculates the expected log-likelihood, which is a concave function when considering the GLM weights, owing to the inclusion of a Bernoulli GLM in the set of functions that transform external inputs into probabilities of HMM emissions[40]. Since there is no closed-form solution for updating the GLM weights, the objective was to compute the global maximum of the expected log-likelihood. To achieve this, the scipy optimize function in Python[82] was used, which employs numerical optimization using the BFGS algorithm[83-86].

By applying the BFGS algorithm, the EM algorithm iteratively refines the GLM weights to maximize the expected log-likelihood,

seeking the optimal parameters that best explain the observed data given the GLM-HMM. This numerical optimization approach enables a more computationally efficient determination of the parameter estimates, leading to improved model performance and accurate representation of the system's dynamics.

So, in this study, the EM algorithm is employed with iterative steps until the difference between consecutive iterations falls below a specified tolerance, ensuring convergence to a stable solution. However, EM algorithms can get stuck in local optima due to their reliance on initial parameter values. To overcome this issue, strategies such as multiple initializations, regularization, and exploration of the parameter space are employed. These enhancements increase the chances of finding favorable solutions. To further validate that the algorithm converges to the global optimum rather than a local optimum, a robust fitting procedure was adopted. The EM algorithm's weights were initialized several times, and for each initialization, the model was fitted. The consistency of the final results across these runs demonstrated that the algorithm consistently converged to the global optimum. Achieving the global optimum of a likelihood function is challenging, and different optimization techniques may be needed depending on the specific problem. Therefore, selecting the right optimization strategies tailored to the problem's characteristics is essential for optimal parameter estimation and model fitting.

Moreover, to assess the uncertainty and estimate the posterior standard deviation of the model weights, the inverse Hessian of the optimized log-posterior was calculated. This computation was performed using Autograd, a powerful Python package and automatic differentiation library that simplifies and enhances gradient-based optimization.

Initialization: The initialization of the GLM-HMM weights followed a specific procedure. The observation weights for the GLM-HMM were initialized using a noisy version of a simple Bernoulli GLM. Initially, a 1-state GLM was fitted, and then the GLM-HMM was initialized with multiple states based on that basic GLM with added noise. This initialization approach provided a reasonable starting point for the GLM-HMM optimization process.

Regarding the transition weights, they were initialized with a vector of zeros, implying no initial knowledge about the transitions between states. Furthermore, to establish prior information for the initialization of the latent states, a uniform distribution prior was set. This choice of prior reflects a neutral assumption, where all states are considered equally probable at the outset.

Due to employing this initialization strategy and setting informative priors, we reached a systematic and principled starting point for the EM algorithm, facilitating more stable and reliable convergence to meaningful solutions during the subsequent parameter estimation and model fitting process.

**GLM-HMM fitting process.** We present the results of applying the GLM-HMM independently to each mouse in Fig. 6. However, establishing a direct relationship between the obtained states across different animals poses challenges. To address this, we adopted a multistep fitting technique. Initially, we combined the data from all mice into a unified dataset.

For the IBL dataset, we aggregated the data from all 37 mice. Employing Maximal Likelihood estimation and the EM algorithm, we analyzed this pooled data using a 1-state GLM-HMM, which represents a simpler GLM. Subsequently, we used the obtained weights from this initial step as the initialization for the GLM weights during the fitting process of a $K$-state GLM-HMM. By doing so, we obtained a global fit by combining the dataset from all mice and fitting a $K$-state GLM-HMM. This approach allows us to assess the relationship between states across different animals and derive a unified representation of the latent states, providing valuable insights into the underlying dynamics observed across the entire dataset.

Following the global fit of the model, we proceeded with an individual fit, resulting in a distinct GLM-HMM fit for each animal. During this individual fit process, we initialized each animal's model using the parameter values obtained from the global fit, which were derived from a model fitted to the pooled data from all mice. To identify the optimal initialization parameters, we conducted 50 initializations and compared the log-likelihood values for the training dataset, selecting the set of parameters that yielded the best fit. We then ran the EM algorithm until convergence for each animal's model.

By employing this individual fitting approach with the best initialization parameters, there was no longer a need to permute the retrieved states of each animal to assign logically similar states to one another. Consequently, the recovered parameters are presented in Fig. 6a, b, illustrating the distinct GLM-HMM fits for each mouse, facilitating an analysis of individual behavioral patterns and underlying dynamics.

In this analysis and fitting process, we incorporated Gaussian noise into the GLM weights, enabling better discernment of the initialized states.

As demonstrated here, the outlined initialization method exhibits sufficient stability and reliability, enabling the recovery of GLM-HMM parameters across a wide range of relevant parameter regimes in our analysis. This approach ensures robust estimation and fosters confidence in the model's ability to accurately capture the underlying dynamics of the system even under diverse conditions.

**Fitting the psychometric curve.** The psychometric function was derived by plotting the percentage of choices made by the animal against various stimulus values. In this paper, the psychometric curves illustrate the animal's rightward choice probability as a function of stimulus intensity and is presented in Fig. 2c. To fit the psychometric curve to the sigmoid function, maximum likelihood estimation was used, resulting in the following formulation:

$$p(\text{choice} = \text{R|SC}) = \frac{1}{1 + \exp(-\mathbf{w}_1^{\text{ob}}\mathbf{x}_1^{\text{ob}})} \quad (33)$$

in which, the SC represents the stimulus contrast. To minimize the loss function, the study employed the Python package 'optimize.minimize' from the scipy library. This package provides numerical optimization tools that enable accurate minimization of the loss function, facilitating the estimation of the model parameters and the fitting of the psychometric curve.

## K-fold cross-validation for GLM-HMM

A cross-validation approach was employed to evaluate the model's performance, involving random splitting of the data into 5 folds for training and testing. Approximately four-fifths of the data sessions were used for training the model, while the remaining randomly selected sessions were held out for testing the fitting performance. The sessions from all participating animals were evenly considered to ensure that mouse-to-mouse variability did not influence the cross-validation process.

This analysis revealed that a GLM-HMM with 3 to 5 latent states yields appropriate log-likelihood estimations. Following the fitting of the model with 1 to 5 states, we computed the log-likelihood of the test data, depicted in Fig. 2a. Also, the test log-likelihood was calculated for all individual fits and is presented in Fig. 6c. During the fitting procedure, the EM algorithm was executed during the training data sessions. Subsequently, in the testing stage, the likelihood of the remaining data was calculated based on the model parameters obtained from the training procedure. The forward pass was performed only once during this stage, and the likelihood of the test data was obtained by summing over all states as $l = \sum_{k=1}^{K} \alpha_{T,k}$, where $\alpha_{T,k}$ was computed solely on the held-out sessions. It is important to note that the GLM-HMM

parameters were solely calculated during the training procedure and used for evaluating the model's performance on unseen data during testing.

In our analysis, Fig. 2, we have calculated the log-likelihood using the described procedure, which is the log of the calculated likelihood and we call it $L = \log(l)$. In some results, we have used the unit term "bits per trial". This $L$ with this unit is acquired by calculating $L$ of the test data on held-out sessions as described, and then subtracting the log-likelihood of the identical dataset considering a Bernoulli model for observed data. This is a baseline model and its corresponding log-likelihood is shown by $L_0$. Then we divided it by $T_t \log(2)$ in which $T_t$ is the test set size (the number of trials). The predictability of test sets can be greatly enhanced by even tiny amounts of log-likelihood when expressed in bits per trial.

In our analysis, we calculated the log-likelihood using the described procedure, denoted as $L = \log(l)$ and in some of the results, we used the unit term "bits per trial" to quantify the log-likelihood.

To obtain the log-likelihood in bits per trial ($L$ bpt), we followed a specific process. Firstly, we computed $L$ for the test data on held-out sessions, as described previously. Then, we determined the log-likelihood of the same dataset under the assumption of a Bernoulli model for observed data and multinomial GLM for transition data, which we refer to as the baseline model. We denoted the baseline model corresponding log-likelihood as $L_0$. The difference between $L$ and $L_0$ represents the enhancement in log-likelihood due to the GLM-HMM performance. Finally, to express this enhancement per trial, we divided it by $T_t \log(2)$, where $T_t$ is the size of the test set (the number of trials).

The resulting value in bits per trial provides a measure of predictability for the test set. Even minute increments in log-likelihood, when expressed in bits per trial, can lead to significant improvements in the predictability of the test data compared to the baseline model. To illustrate this, consider a numerical example where the log-likelihood difference estimate is 0.02 bpt. This implies that the test data is approximately 1047128.54 times more likely to have originated from the GLM-HMM compared to the baseline model when the test data comprises 1000 trials.

This representation in bits per trial, referred to as NLL, allows for a more intuitive understanding of the model's performance and enables meaningful comparisons between different models and dataset. So we can write it as:

$$L_{\text{bpt}} = (L_t - L_0)/(T_t \log(2)) \tag{34}$$

in which bpt is an abbreviation for bit per trial unit.

### Synthetic data
We simulated data (Fig. 7d) using the best-fitting model for each individual animal from the IBL dataset. For each animal, we reconstructed the model and generated simulated data on a session-by-session basis. In each session, we used the true stimulus values presented to the animal and generated corresponding choices and latent states, along with the covariate values, following the model structure. Specifically, for each trial, latent states were sampled using the transition model-either a fixed transition matrix (model without GLM-T) or a multinomial GLM with input-driven transitions (model with GLM-T). Conditioned on the latent state, choices were then sampled from a Bernoulli GLM. This process ensured that trial-by-trial choices and latent states were newly simulated while preserving the structure of the task, enabling the use of ground-truth labels for model recovery evaluation. We simulated data with $K = 4$ latent states, using the original session lengths and recursively updating the covariates to reflect the simulated trial histories. The observation

model was a Bernoulli GLM with covariates including past choice, stimulus side, and previous reward. These covariates were updated trial-by-trial based on the simulated choice and feedback (e.g., reward outcome and correct side). Simulations were run with fixed random seeds to ensure reproducibility. For the model without GLM-T, transitions were static and governed by a fixed transition matrix without covariate inputs. In contrast, the model with GLM-T had a multinomial GLM with input-driven transitions, where transition probabilities were modulated by exponentially filtered versions of the different behavioral covariates.

### Reporting summary
Further information on research design is available in the Nature Portfolio Reporting Summary linked to this article.

### Data availability
In this study, we assessed behavioral data from 123 mice in the IBL decision-making task, analyzing all sessions from a subset of 37 mice to ensure generalizability. These 37 mice were selected based on specific inclusion criteria: they had to have completed a minimum of 30 sessions, and we prioritized sessions with a low number of error trials, where the animal either failed to make a choice or timed out. We studied the stationary behavior phase after the training period, considering both biased and unbiased data blocks. In unbiased blocks, the stimulus appeared equally on both sides at the start of each session, while in biased blocks, it appeared with an unequal probability, with block lengths ranging from 20 to 100. The dataset is publicly available at https://doi.org/10.6084/m9.figshare.11636748. For efficient data access, please see https://int-brain-lab.github.io/iblenv/notebooks_external/data_download.html and use the data tag "2023_Q1_Mohammadi_et_al". Detailed explanations can be found in the paper's GitHub repository. We applied our model to this dataset; further information about the data can be found in ref. 87. The dataset includes both male and female mice; however, sex was not analyzed as a variable, as it was not the focus of this modeling study. Source data are provided with this paper.

### Code availability
The GLM-HMM code package developed for this paper, as well as the figures presented, can be accessed at https://github.com/Zeinab-Mohammadi/glm-hmm_final.git. We used and extended the Bayesian SSM framework[88] by adding new functionality to the GLM-HMM and providing a code script. This code base was subsequently used for performing model inference in this manuscript. To access the code, please refer to our modified version of the SSM package available at https://github.com/Zeinab-Mohammadi/ssm. For a practical demonstration of the package's application in our analysis, including a generative model and its parameters recovery process, see https://github.com/Zeinab-Mohammadi/ssm/blob/master/notebooks/2c-Input-Driven-Transitions-and-Observations-GLM-HMM.ipynb. Additional details can be found in https://github.com/Zeinab-Mohammadi/ssm/blob/master/ssm/hmm_TO.py.

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

## Acknowledgements
We extend our sincere appreciation to S. Linderman and his group for the outstanding Bayesian Learning and Inference for State Space Models (SSM) package, from which our code in this paper is derived. We would like to acknowledge the feedback received from the International Brain Laboratory (IBL) and the Theory Working Group throughout the course of this research project. Special thanks go to our board members, L. Paninski, A. Pouget and I. B. Witten, for their constructive input on this manuscript. We are also grateful to H. Bayer and the IBL staff for their support. We express our gratitude to the members of the Laboratory of J. W. Pillow, especially M. Creamer. Portions of the text were edited with the assistance of a large language model (ChatGPT) to improve clarity and grammar. The authors reviewed and verified all content. This work was supported by grants from the Simons Collaboration on the Global Brain (SCGB AWD543027 to J.W.P.), the National Institutes of Health BRAIN initiative (9R01DA056404-04 to J.W.P.), and a U19 NIH-NINDS BRAIN Initiative Award (5U19NS104648 to J.W.P.). Z.M. was supported by a Swartz Foundation Postdoctoral Fellowship.

## Author contributions
Z.M. developed the model and analyzed the data under J.W.P.'s guidance. Z.M. wrote the original draft and generated the figures. Z.M. and J.W.P. interpreted the results and edited the manuscript. Z.C.A. and J.W.P. provided technical and analysis support and reviewed the paper. The IBL conducted the experiment and collected the data.

## Competing interests
The authors declare no competing interests.

## Additional information

## The International Brain Laboratory

**Zeinab Mohammadi** [1] ✉, **Zoe C. Ashwood** [1,2] & **Jonathan W. Pillow** [1]

