## [Transparent Peer Review file · Nature Communications]

Identifying the factors governing internal state switches during nonstationary sensory decision-making

Corresponding Author: Dr Zeinab Mohammadi

Version 0:

Reviewer comments:

Reviewer #1

(Remarks to the Author)

This manuscript by Mohammadi et al. is an interesting and needed generalization of the GLM-HMM framework, which has become a useful tool for analyzing diverse internal states during decision making. It extends the “standard” GLM-HMM proposed by Ashwood et al. (2022) by incorporating non-stationary transition probabilities among the assumed states. More specifically, the extended GLM-HMM infers dynamically changing transition probabilities that depend on prior history, suggesting that variables such as choice and reward history can have significant impact on state transitions. One related notable finding is that higher reward rate is associated with transition into “biased” or disengaged states.

Overall, the extended model significantly improves the fit of choice behavior over the previous model by combining Bernoulli and multinomial GLM in a novel way. However, the study lacks clarity on what additional insights it offers beyond the previous standard GLM-HMM, aside from simply providing a better fit to the behavioral data. In other words, we believe there is still room to better demonstrate the full utility of the presented/extended model. The transition GLM (“GLM-T”) weights are presented, but are not sufficient in illustrating how they impact the choice behavior overall. The study presents evidence on how prior trial history affects state transitions, but does not explore the reverse: how the newly inferred states, which are presumably more accurate, provide new insights into the behavior they produce.

One possible analysis is to compare how the observation GLM weights (“GLM-O,” Fig. 2b) differ when GLM-T is excluded or included. Another crucial piece of information missing from the manuscript is the analysis of behavioral correlates for each state estimated with the GLM-T. For example, We wonder if the inclusion of GLM-T enables better distinction between states in terms of performance (probability correct), reaction time, and actual rewards collected. The authors suggest the link between disengagement and satiety from transition weights alone, but this could be more exhaustively tested; for example, by looking at whether occupancy of those states also tend to increase toward the end of the session (when mice have accumulated overall more rewards over the experiment). Also we expect that (filtered) reward rate is correlated with probability of occupying biased (disengaged) states, but this has not been explicitly tested. These analyses could make the main claim of the study more convincing.

In addition, there are certain places where the assumptions of the extended GLM-HMM model need to be checked. One issue is the question of how it truly differentiates from the previous GLM-HMM model with fixed transition probabilities. Although transition probabilities are fixed for the standard GLM-HMM, there might be scenarios where it's difficult to distinguish it from non-fixed cases, since the posterior state probabilities are after all estimated on a trial-by-trial basis. This raises the possibility that if transition probabilities (for the extended model) are not dynamic enough, the resulting behavior might not be so distinguishable from the standard model. We suggest that authors should find a way to directly address this issue, perhaps through model recovery from simulated data. Relatedly, could authors plot how the (inferred) transition probabilities change over time, instead of their averaged values (Fig.3e), to connect with the information shown in Fig.5a more intuitively?

Lastly, if we are not mistaken, the covariates that are included for the transition GLM are assumptions that were pre-selected for the model rather than those that survive among other candidate covariates. Could the authors provide a rationale for selecting these covariates and not others? And could they comment on whether there might be other potential covariates that were not tested here but could be important?

Minor comments

Either Fig. 3f or Fig. 3g for the point corresponding to 5-state & Tau=8 seems to be incorrect. We assume these two plots should be different representations of the same data. In Fig. 3f for 5-state case, Tau = 8 (green) has slightly higher LL than Tau = 2 (purple). But looking at Fig. 3g 5-state graph (in red), Tau = 2 has a higher LL than Tau = 8.

In Fig.6b, the X-axis has 6 points but the labels only have five. We were wondering if these were labeled incorrectly because “filtered reward” for biased states are not predominantly positive.

Fig. 3d (transition bias) is not mentioned anywhere in the main text and could use some explanations (the same for Fig. S6).

Fig.4a is somewhat confusing. We suggest adding axis labels in each panel and clearly marking them as “example session 1”, “example session 2”, etc.

“Stimulus side,” as explained in the manuscript (#124-125) is equivalent to what’s commonly known as “win-stay/lose-switch.” We were wondering why the former term was adopted and not just called the latter. For this variable, isn’t there no relation with specific stimulus side as the mice can WS-LS on either side?

In line #367-368, “solid black line” should be “solid red line” according to Fig. 6c figure caption.

Line #388-389 states: “For the transition model, filtered choice, stimulus side, and reward were the most influential covariates, ...” Related to the comment above, this statement could be toned down a little because these factors are assumptions that were pre-selected for the model rather than those that survive among other candidate covariates.

The Methods section (Section 5) for describing the dataset and the exclusion criteria could be a little more detailed. It seems like the current study is selecting the same 37 mice (out of 123) that were used in the previous study by Ashwood et al. (2022), which we found out only after referring to it for more information. The authors could either explicitly state this or make the Methods more self-contained, since the readers might not be aware of the previous work.

References

Le et al., which is cited in #36, has been published in PLOS Comp. Bio.:

Le, N. M., Yildirim, M., Wang, Y., Sugihara, H., Jazayeri, M., & Sur, M. (2023). Mixtures of strategies underlie rodent behavior during reversal learning. *PLOS Computational Biology*, 19(9), e1011430.

Similar to above work, it might be worth also citing the following study, which applies HMM to decision making:

Li, J. J., Shi, C., Li, L., & Collins, A. G. (2024). Dynamic noise estimation: A generalized method for modeling noise fluctuations in decision-making. *Journal of Mathematical Psychology*, 119, 102842.

(Remarks on code availability)

Reviewer #2

(Remarks to the Author)

In the present manuscript, the authors present an adapted hidden Markov model that allows for various covariates from the experiment to guide the transitions between decision making states, and (potentially) separate covariates to guide the decision making ‘policy’ within those different states. They then apply their model to some rat perceptual learning data in which the probability of left or right decisions (i.e., a bias) should increase or decrease across blocks of trials. They find some apparent evidence of a four state model in which left or right bias appears to be crossed with engaged or not (so engaged left, engaged right, left bias, right bias).

I think this is an interesting manuscript in that it develops a useful statistical model to capture important trends in the decision making style across trials. These HMMs with transitions across decision states are conceptually easy but very difficult to fit to data. Adding to this complexity is the possibility of four different sets of GLM weights to predict choices. Finally, the authors use a nice data set to illustrate the utility of their model – not only is there left and right biases, but these are very long experiments that would almost certainly catch the rat off task from time to time (and as their model suggests, they do engage in off task behavior).

All of that said, the novelty of the research seems to be a bit low for this particular outlet in my opinion. According to the authors, this division between two separate generalized linear model structures to guide both the transition between the states and the decisions within those states is the main novelty of their research. To me, the main contribution of their modeling efforts is that they allow for two very general (literally GLM) structures to modulate the two main components of their model, one being a transition among the states and one being the decisions within those states. While this is interesting, methodologically speaking this is not a significant leap forward. Furthermore, the evidence for a need of a separate transition GLM seems quite low. Maybe it is just a scaling issue, but Figure 2a is not really convincing that having a transition model is actually necessary.

Although the results are also interesting, it is perhaps not that surprising that they would find a model that supports the two real conditions of the experiment (left or right bias) crossed with ‘engaged’ or ‘not engaged’. To be honest, it was a little unclear what the inputs were to the GLM observation, despite a whole section for it in 4.5. For example, they say that ‘bias’ was in the set of observational covariates, but it is not defined in 4.5. It is hard to understand how a variable called ‘bias’ could meaningfully contribute to the fits if there are already regressors for the stimulus itself and the past choice and previous

stimulus. In any case, I would worry that they are finding evidence for a four state model simply because the models that explain decisions are far too simplistic. It is possible that a more detailed model of the decision itself might imply that there are actually fewer states. For example, potentially even having a regressor for 'number of trials performed in this block' might soak up a lot of within-state variability to remove the need for two of those four states. Because questions like this were not really fully examined, the results surrounding the transition GLM's contribution are pretty minimal. However, even if they were significant, it is not clear that an incremental methodological advancement in a well established field of HMMing makes sense at this particular outlet.

(Remarks on code availability)

Reviewer #3

(Remarks to the Author)

The authors investigated behavioral strategies used by mice performing a binary decision-making task with a time-varying distribution over correct choices. To do so, they fit two sets of GLMs: one that predicts observed choices as a function of sensory inputs, task covariates, and latent state, and another that predicts transition probabilities to each latent state. Crucially, this setup allows the transition probabilities to vary over time, in contrast with traditional HMMs, and thus is a sensible choice for analyzing behavior in a non-stationary environment. In sum, they find four interpretable states that govern behavior during this task, which are broken down into "engaged" states, in which animals appear to rely on the observed stimulus, and "biased" states, in which animal's choices seem heavily biased towards one direction regardless of stimulus. They characterize various task-relevant factors that influence transitions between these states, and provide a detailed account of how transition rates are modulated over time by the underlying evolving task structure.

Strengths:

The authors do an excellent job demonstrating that the behavioral data is well-explained by their proposed four-state model, and their interpretation of these states is solidly supported by breakdowns of GLM weights and psychometric curves.

Concerns/Questions:

The relationship between this work and the cited "Extracting the dynamics of behavior in sensory decision-making experiments" (Roy et al., 2021, Neuron) should be explicitly discussed. In particular, both make inferences about animal behavior from the same dataset, and both employ a fitting method that permits a time-varying observation model. The authors should make an effort to underscore what findings are uniquely highlighted by their GLM-HMM approach that couldn't be captured by the model used there.

I am somewhat puzzled by why state transition probabilities are modeled by the transition GLM in the way they are. If I am understanding correctly, only the probabilities of transitioning to a particular latent state are parameterized, regardless of starting latent state, for a total of K transition probabilities (this interpretation is consistent with the exposition in the text as well as Fig. 5). This is quite a restrictive choice compared to standard HMMs, which parameterize all K^2 pairs of from-and-to. In this case, I would expect that transitions "Engaged R \rightarrow Biased R" would be significantly more probable than "Engaged L \rightarrow Biased R". Indeed, other works that use time-varying transition probabilities, such as the cited work Calhoun et al. 2019, fit a separate GLM for the transition probabilities out of each latent state. I would like to see that the conclusions drawn are not overly dependent on this particular modeling choice.

While directions for future work are discussed thoroughly, interpretations and caveats of the presented findings seemed a tad scarce in the discussion. For example, the authors allude to occupancy of disengaged states being related to satiety. The authors could expand on this, possibly by commenting on the robustness of this conclusion, especially when considering that the delineation appears less clear for the 5 state model as shown in Fig. S2b.

Minor/miscellaneous:

Is there a relationship between the model's predicted probability of the animal being in a biased/disengaged state during some time window and the number of incorrect choices observed? The authors state that performance remains high even in the biased states because of imbalanced prior probabilities, but I don't see this quantified anywhere. Reporting a prevalence-independent performance metric like balanced accuracy could also be insightful here.

(Remarks on code availability)

Reviewer #4

(Remarks to the Author)

(Remarks on code availability)

Version 1:

Reviewer comments:

Reviewer #1

(Remarks to the Author)

We would like to thank the authors for thoughtfully and thoroughly addressing our concerns. Below, we have outlined a few minor final points.

Rebuttal point 3:

"... This result highlights that our model, which integrates both GLM-T and GLM-O, governs how behavioral states evolve over time while capturing choice behavior patterns, whereas GLM-O alone defines within-state choice behavior. The consistency of GLM-O weights across models further supports the idea that the transition process (GLM-T) does not interfere with moment-to-moment decision weightings but instead regulates long-term behavioral shifts. We have added this plot to the supplementary materials."

Thank you for running this additional analysis. We now see that the GLM-O weights do not really differ between the models with or without the GLM-T component. We were wondering which part of the main text (if any) refers to these results. It might be worth explaining this point in a little more detail, because it was at first counterintuitive that the two weights should not influence each other.

Rebuttal point 4:

Thank you for following up with a comprehensive set of analyses. The result suggests that including GLM-T component leads to higher confidence in state assignments with distinct predictions for behavior such as RT and switching. The new figures were overall clear, with the exception of Fig.9c: the plots were too cluttered and the analysis here seems to rely on qualitative judgment. Could this be replaced, for example, with the entropy of the $P(\text{state})$ or showing just for the maximum probability instead, to avoid redundancy with panels (a)-(b)? Also we assume this is for an example session. The figure is also missing the legend for the colors of the plots.

Rebuttal point 5:

Thank you for testing this idea. We think the results are clear.

Rebuttal point 6:

Thank you for providing these extensive simulation results. The model recovery results and Fig.S10 were clear in demonstrating the utility of the transition model. We would just recommend clarifying the simulation conditions (e.g., parameters) for generating the synthetic data from two models in Methods, if this has not already been done.

Rebuttal point 7:

Thank you for the additional explanation.

(Remarks on code availability)

Reviewer #2

(Remarks to the Author)

I don't really have any further comments. Although I appreciate the authors' efforts, the model they use here is more of a statistical approach (through the use of very general GLMs) and lacks a lot of theoretical interest, which is the type of approach I usually gravitate toward. The authors did a lot to justify their position, and while I appreciate their efforts, the manuscript remains in a similar state as the first version although some of the additional analyses are slightly more convincing statistically, it is still not clear to me whether the results are sort of guaranteed from the structure of the task. We continue to disagree about the novelty of their approach for this particular high-impact journal, but it is not up for me to decide so hopefully the AE can make a decision here.

(Remarks on code availability)

Reviewer #3

(Remarks to the Author)

I thank the authors for their additional analyses and for the expanded discussions in response to my points. I have one remaining comment.

Re: Response to "I am somewhat puzzled by why state transition probabilities are modeled by the transition GLM in..."

If a separate multinomial GLM is indeed being fit for each latent state, I would interpret that as meaning a separate set of K weight vectors is used for modeling the transition probabilities of leaving each state. However, Based on equation (2) and lines 90-94, as well as equation (12) and lines 607-611, the multinomial GLM used for state transitions parameterizes probabilities of transitioning to states with just one set of K weight vectors. What is still not clear here is how these probabilities take into account the starting state from which the state transition is occurring. Is starting state information somehow implicitly embedded in x_t^{tr} ? If not, then the problem of indistinguishability of $P(\text{Engaged R} \rightarrow \text{Biased R})$ vs $P(\text{Engaged L} \rightarrow \text{Biased R})$ remains.

This point remains my biggest concern. If K separate multinomial GLMs are being fit for transition probabilities, the quoted equations/text should be updated to clarify this.

(Remarks on code availability)

Reviewer #4

(Remarks to the Author)

(Remarks on code availability)

Version 2:

Reviewer comments:

Reviewer #1

(Remarks to the Author)

The authors have adequately addressed our questions and comments, and we have no further remarks.

(Remarks on code availability)

I did not test the code but the it seems well documented.

Reviewer #3

(Remarks to the Author)

Thank you for your response. I don't have any more comments or questions.

(Remarks on code availability)

Dear Reviewers,

We sincerely appreciate your time and effort in reviewing our manuscript. Below, we have provided detailed responses to each of your queries and comments, addressing them point by point. Your constructive feedback has been invaluable in improving the clarity and rigor of our work.

Reviewer #1 (Remarks to the Author):

1. This manuscript by Mohammadi et al. is an interesting and needed generalization of the GLM-HMM framework, which has become a useful tool for analyzing diverse internal states during decision making. It extends the “standard” GLM-HMM proposed by Ashwood et al. (2022) by incorporating non-stationary transition probabilities among the assumed states. More specifically, the extended GLM-HMM infers dynamically changing transition probabilities that depend on prior history, suggesting that variables such as choice and reward history can have significant impact on state transitions. One related notable finding is that higher reward rate is associated with transition into “biased” or disengaged states.

We thank the reviewer for this positive assessment of our manuscript’s contributions.

2. Overall, the extended model significantly improves the fit of choice behavior over the previous model by combining Bernoulli and multinomial GLM in a novel way. However, the study lacks clarity on what additional insights it offers beyond the previous standard GLM-HMM, aside from simply providing a better fit to the behavioral data. In other words, we believe there is still room to better demonstrate the full utility of the presented/extended model.

We thank the reviewer for these insightful comments and appreciate their recognition of our model’s improvements. We agree that it is important to highlight the additional insights our model provides. Below, we describe the additional analyses and figures we have included to address the reviewer’s specific suggestions.

3. The transition GLM (“GLM-T”) weights are presented, but are not sufficient in illustrating how they impact the choice behavior overall. One possible analysis is to compare how the observation GLM weights (“GLM-O,” Fig. 2b) differ when GLM-T is excluded or included.

Thanks for this suggestion. We have now added plots comparing the GLM-O weights obtained from models with and without GLM-T. These results demonstrate that including the GLM-T model does not significantly alter the GLM-O weights, confirming that the two components operate independently. This aligns with our expectations, as GLM-T is not designed to modify observation weights or choice behavior substantially.

However, the addition of GLM-T enhances the model’s ability to capture behavioral dynamics, as reflected in the higher log-likelihood (LL) value in the 4-state model compared to the 3-state model used in Ashwood et al. (2022). Notably, Ashwood et al. (2022) model only included

GLM-O weights for the unbiased part of the data (the first 90 trials of each session), where environmental dynamics were not present. In contrast, our 4-state model, which explicitly incorporates bias values, provides a more accurate description of the changing environment, suggesting that state transitions play a key role in behavioral adaptation.

This result highlights that our model, which integrates both GLM-T and GLM-O, governs how behavioral states evolve over time while capturing choice behavior patterns, whereas GLM-O alone defines within-state choice behavior. The consistency of GLM-O weights across models further supports the idea that the transition process (GLM-T) does not interfere with moment-to-moment decision weightings but instead regulates long-term behavioral shifts. We have added this plot to the supplementary materials.

4. The study presents evidence on how prior trial history affects state transitions, but does not explore the reverse: how the newly inferred states, which are presumably more accurate, provide new insights into the behavior they produce.

Another crucial piece of information missing from the manuscript is the analysis of behavioral correlates for each state estimated with the GLM-T. For example, We wonder if the inclusion of GLM-T enables better distinction between states in terms of performance (probability correct), reaction time, and actual rewards collected.

Thank you for your thoughtful feedback and for highlighting the importance of analyzing behavioral correlates of the inferred states.

We compared the performance of both models. The left scatter plot and the right density plot both show that states inferred under the model with GLM-T generally have higher probabilities compared to those from the standard GLM-HMM. The upward shift of many points above the diagonal ($y = x$) indicates that, for the same states, the model with GLM-T assigns a higher probability. This suggests that the inclusion of GLM-T leads to more confident state assignments and better performance. The density plot further supports this pattern, showing that the most frequent state probabilities tend to be higher under GLM-T. These findings indicate that the model with GLM-T provides more confident state estimations.

Furthermore, to address the reviewer’s point about how the newly inferred states provide insights into behavior, we plotted the posterior state probabilities across trials for the standard GLM-HMM (top), the model with GLM-T (middle), and their combined visualization (bottom). These plots show that incorporating GLM-T results in higher confidence in state assignments. In the model without GLM-T, multiple states often have moderate probabilities, leading to greater uncertainty in state identity. In contrast, with GLM-T, states reach higher probabilities, reducing overlap and making transitions between them more distinct.

Additionally, the model with GLM-T better captures transition patterns that align with the task structure, particularly biased block switching (e.g. left-biased states appear more frequently in blocks where the stimulus is more probable on the left side).

Overall, the model with GLM-T provides greater confidence in state estimations—not only because state probabilities are generally higher, but also because it more successfully captures the bias-block structure and accurately tracks switches between them. For example, as shown in the lowest panel with green squares, the states inferred by the model with GLM-T (solid lines) consistently align with the expected transitions, further demonstrating its improved ability to reflect behavioral adaptations to stimulus probability changes.

On the other hand, below we have plotted the reward rate for both the model with GLM-T and the model without GLM-T (engaged states in green and disengaged states in orange). Since the observation model parameters remained largely unchanged between the two models, the reward rate also showed minimal differences. This suggests that the inclusion of GLM-T did not significantly alter overall reward outcomes, reinforcing the expectation that the estimated states in our model do not strongly influence choice behavior.

Furthermore, we analyzed response times instead of reaction times here, as reaction time data was unavailable for most IBL data sessions. Response time is defined as the interval between stimulus onset and the moment the mouse receives feedback on its decision. In contrast, reaction time in the IBL task refers to the time from stimulus onset to the mouse's initial movement of the wheel. While response times varied widely, ranging from a few hundred milliseconds to as long as 10 seconds in extreme cases, typical reaction times (estimated from a limited number of sessions with reliable data) were consistently in the range of a few hundred milliseconds. Compared to other tasks, the IBL task is characterized by relatively short reaction times.

We analyzed the difference in response times between disengaged and engaged states across all animals. Specifically, we computed the difference in the 90th percentile response time for each state in every mouse and plotted the results, including 95% bootstrap confidence intervals ($n = 2,000$). This analysis allows us to quantify how distinct these behavioral states are in terms of response time.

To further assess the impact of GLM-T, we compared this difference across models with and without GLM-T. As shown in the below figure, for all animals, the difference in 90th percentile response time between disengaged and engaged states was larger in the model with GLM-T. This suggests that the inclusion of GLM-T enhances the distinction between these states, making disengaged trials more clearly separated from engaged trials in terms of response latency. The larger response time gap under model with GLM-T indicates that this model better captures behavioral differences associated with engagement and disengagement.

To better address the reviewer’s question regarding the performance of the models, we plotted the fractional occupancy of the four discrete states in the model without GLM-T for all IBL data, similar to Fig. 4c. This analysis was conducted across all trials, covering both right-biased and left-biased blocks. The results are displayed separately for each block type. As seen in the plot, the background stimulus bias was not captured by the model without GLM-T.

Furthermore, to evaluate performance in relation to switching patterns in both models, we analyzed the switching data. The below figure compares the first switching probability across different trials, representing the time it takes for animals to switch states after encountering a

new biased block of data. The results are shown for models without GLM-T (left) and with GLM-T (right). The model without GLM-T shows no specific peak, indicating that it fails to capture structured state transitions following block changes. The model with GLM-T exhibits a distinct peak around trials 6-8, indicating that animals tend to switch states after detecting a block transition, with a slight delay. This suggests that incorporating GLM-T allows the model to account for the bias block, understand it, and adjust the states accordingly with a short delay, so that it better aligns with behavioral dynamics, as animals appear to systematically adjust their state based on changes in stimulus blocks. These findings further support the idea that GLM-T enhances the model's ability to capture meaningful behavioral patterns, making it a more behaviorally relevant and interpretable framework for decision-making in dynamic environments.

We have included these figures in either the paper or the supplement.

5. The authors suggest the link between disengagement and satiety from transition weights alone, but this could be more exhaustively tested; for example, by looking at whether occupancy of those states also tend to increase toward the end of the session (when mice have accumulated overall more rewards over the experiment) . Also we expect that (filtered) reward rate is correlated with probability of occupying biased (disengaged) states, but this has not been explicitly tested. These analyses could make the main claim of the study more convincing.

Thank you for these suggestions to clarify the relationship between disengagement and satiety. As the reviewer suggested, we examined the relationship between them by plotting the probability of occupying engaged and disengaged states throughout the session. We interpolated 100 points from each session and then averaged them across all sessions for all animals. The plot shows that at the beginning, there is a clear warm-up effect, where mice transition from disengaged to engaged states, indicating an initial adjustment phase before stable task performance. Toward the end of the session, the occupancy of disengaged states increases while engaged states decrease, supporting the idea that as mice accumulate more rewards, they become less engaged, potentially due to satiety. This provides initial evidence for the proposed link between disengagement and satiety.

Furthermore, to study the relationship between disengagement and satiety, we generated two analysis plots. The first figure shows the relationship between filtered reward (gray) and the probability of being in the disengaged state (orange) across trials. The shaded regions represent different biased blocks throughout the session. For most trials, as the filtered reward rate increases, disengagement probability also starts to increase, suggesting a direct correlation between the two.

The second figure further confirms this relationship by plotting $P(\text{disengaged})$ vs. filtered reward across trials. The red trend line shows a positive correlation. So we found a statistically significant positive moderate correlation between the filtered reward rate and the probability of disengagement ($r \approx 0.30$, $p < 0.001$), where $r = 0.30$ (Pearson correlation coefficient) indicates a moderate positive correlation, and $p < 0.001$ (p-value) confirms that this relationship is highly unlikely to be due to chance. This moderate correlation suggests that as the reward rate increases, disengagement probability also tends to increase, but other factors likely contribute as well, supporting the idea that reward rate plays a role in disengagement dynamics.

6. In addition, there are certain places where the assumptions of the extended GLM-HMM model need to be checked. One issue is the question of how it truly differentiates from the previous GLM-HMM model with fixed transition probabilities. Although transition probabilities are fixed for the standard GLM-HMM, there might be scenarios where it's difficult to distinguish it from non-fixed cases, since the posterior state probabilities are after all estimated on a trial-by-trial basis. This raises the possibility that if transition probabilities (for the extended model) are not dynamic enough, the resulting behavior might not be so distinguishable from the standard model. We suggest that authors should find a way to directly address this issue, perhaps through model recovery from simulated data. Relatedly, could authors plot how the (inferred) transition probabilities change over time, instead of their averaged values (Fig.3e), to connect with the information shown in Fig.5a more intuitively?

Also, to address the reviewer's point regarding how the extended GLM-HMM differentiates from the standard model, please see the figure below, which is a supplementary figure (Figure S5) from the paper employing synthetic data. This figure illustrates the fractional occupancy of different states for models with and without GLM-T, separated by left-biased and right-biased task conditions. The comparison shows that when GLM-T is included (panel a), the state occupancy distributions adapt more distinctly to task biases, whereas without GLM-T (panel b), the state occupancies appear without being affected by data bias blocks. This further supports that the inclusion of trial-dependent transitions allows the model to capture dynamic shifts in behavioral states, distinguishing it from the standard GLM-HMM with fixed transition probabilities.

To address the reviewer's question, we conducted two model comparisons between models with and without GLM-T using two types of synthetic data.

- First, we simulated data from a model without GLM-T (4-state) and fitted the GLM-T and non-GLM-T models to this data (left side of the figure below). In this case, both models performed almost similarly.
- Second, we simulated data from a model with GLM-T (4-state) and then fitted both the GLM-T and non-GLM-T models to this data (right side of the figure below). The test log-likelihood comparison shows that the model with GLM-T performs better, achieving an improvement of 0.009 bits per trial over the non-GLM-T model. This demonstrates that when the true generative process involves trial-dependent transitions, the model with GLM-T can better capture behavioral dynamics.

This improvement of 0.009 bits per trial when the data has trial-dependent transitions means that for a dataset with 5000 trials, the data would be approximately 3.52×10^{13} times more probable under the GLM-T model than under the non-GLM-T model.

The difference in performance between these two cases highlights the impact of capturing dynamics effectively. This further highlights the substantial advantage of the GLM-T model in capturing trial-dependent dynamics, especially in behavioral data, where datasets typically consist of a large number of trials.

Per the reviewer's request, we plotted the transition probabilities (instead of their averaged values (Fig.3e)) for both models—with and without GLM-T—to highlight their differences over time.

As shown in this plot, when GLM-T is included, transition probabilities exhibit trial-by-trial fluctuations, adapting dynamically across different states and conditions. In contrast, without GLM-T, transition probabilities remain fixed, lacking the flexibility to adjust to behavioral shifts.

This visualization confirms that the extended model (GLM-T) captures dynamic transitions, differentiating it from the standard GLM-HMM, where transitions are static and do not evolve with trial history or task conditions. Also, the main diagonal plots (high-probability states) follow a similar pattern to Fig. 5a (top session) and are connected to it.

We have included these figures in either the paper or the supplement.

7. Lastly, if we are not mistaken, the covariates that are included for the transition GLM are assumptions that were pre-selected for the model rather than those that survive among other candidate covariates. Could the authors provide a rationale for selecting these covariates and not others? And could they comment on whether there might be other potential covariates that were not tested here but could be important?

We sincerely appreciate the reviewer's insightful comment. In selecting the covariates for the transition GLM, we aimed to include those that based on both our reasoning and the literature, would have the most significant influence on making decision and strategy switching. Specifically, we incorporated history-dependent factors such as the animal's choice history, stimulus side history, and reward history, as these are well-established determinants of decision-making strategies. Additionally, we accounted for warm-up effects using basis functions to capture early-session adaptation in transition probabilities. To ensure that our model appropriately captured the temporal dynamics of these covariates, we searched for the optimal time constant that best described their influence on transitions (Figures 3f and 3g). This involved filtering the covariates with different time constants and selecting the one that most accurately captured the relevant history and period of effect. Even this process required significant time and computational resources due to the large dataset and the complexity of the model. Given these constraints, our approach focused on refining a predefined set of covariates rather than conducting an exhaustive search over a broader range of predictors.

That said, we acknowledge that additional covariates could be relevant to strategy switching. For example, response time could reflect decision urgency or hesitation, movement vigor (e.g., velocity, acceleration) might indicate engagement or motivation, and inter-trial interval could capture the influence of short-term breaks on strategy shifts. Additionally, neural activity in relevant brain regions, if available, could provide insight into strategy adaptation mechanisms, while task engagement metrics (e.g., number of licks, pupil dilation) may indicate attentional fluctuations affecting transitions. Exploring a broader set of covariates and implementing a more data-driven selection approach would be a valuable direction for future work. We have added this to the future work section of the paper in orange as follows: **“On the other hand, it would be useful to explore other covariates that may influence strategy switching but were not included in our current model. While we selected covariates based on prior literature and theoretical reasoning, a more data-driven approach could help identify additional predictors, such as response time, movement vigor, or task engagement metrics, to capture decision-making dynamics in more detail.”**

Minor comments

Either Fig. 3f or Fig. 3g for the point corresponding to 5-state & Tau=8 seems to be incorrect. We assume these two plots should be different representations of the same data. In Fig. 3f for 5-state case, Tau = 8 (green) has slightly higher LL than Tau = 2 (purple). But looking at Fig. 3g 5-state graph (in red), Tau = 2 has a higher LL than Tau = 8.

We sincerely appreciate the reviewer's careful examination of our figures and for catching this discrepancy. We have now corrected the figure to ensure that both panels accurately reflect the same underlying data.

In Fig.6b, the X-axis has 6 points but the labels only have five. We were wondering if these were labeled incorrectly because "filtered reward" for biased states are not predominantly positive.

We have rotated the x-axis labels appropriately, so they are now correctly displayed.

Fig. 3d (transition bias) is not mentioned anywhere in the main text and could use some explanations (the same for Fig. S6).

We explained both in the paper.

Fig.4a is somewhat confusing. We suggest adding axis labels in each panel and clearly marking them as "example session 1", "example session 2", etc.

We edited the labels to make the figure clearer.

"Stimulus side," as explained in the manuscript (#124-125) is equivalent to what's commonly known as "win-stay/lose-switch." We were wondering why the former term was adopted and not just called the latter. For this variable, isn't there no relation with specific stimulus side as the mice can WS-LS on either side?

We thought that win-stay/lose-switch referred to the strategy itself, but "Stimulus side" is actually the name of the regressor that influences it. If the weight was negative, the strategy would instead be win-switch/lose-stay. So we have chosen 'Stimulus side' as the name of the regressor.

In line #367-368, "solid black line" should be "solid red line" according to Fig. 6c figure caption.

Good catch! We just edited the words.

Line #388-389 states: "For the transition model, filtered choice, stimulus side, and reward were the most influential covariates," Related to the comment above, this statement could be toned down a little because these factors are assumptions that were pre-selected for the model rather than those that survive among other candidate covariates.

Thank you for the suggestion—we have revised the statement to clarify this.

The Methods section (Section 5) for describing the dataset and the exclusion criteria could be a little more detailed. It seems like the current study is selecting the same 37 mice (out of 123) that were used in the previous study by Ashwood et al. (2022), which we found out only after referring to it for more information. The authors could either explicitly state this or make the Methods more self-contained, since the readers might not be aware of the previous work.

Thanks, we have added the explanation to Section 5 to make the dataset selection criteria more explicit and self-contained.

Le et al., which is cited in #36, has been published in PLOS Comp. Bio.:

Le, N. M., Yildirim, M., Wang, Y., Sugihara, H., Jazayeri, M., & Sur, M. (2023). Mixtures of strategies underlie rodent behavior during reversal learning. *PLOS Computational Biology*, 19(9), e1011430.

Similar to above work, it might be worth also citing the following study, which applies HMM to decision making:

Li, J. J., Shi, C., Li, L., & Collins, A. G. (2024). Dynamic noise estimation: A generalized method for modeling noise fluctuations in decision-making. *Journal of Mathematical Psychology*, 119, 102842.

Thanks for pointing this out. We have edited reference number 36 and added the recommended related reference (number 37).

Reviewer #2 (Remarks to the Author):

In the present manuscript, the authors present an adapted hidden Markov model that allows for various covariates from the experiment to guide the transitions between decision making states, and (potentially) separate covariates to guide the decision making 'policy' within those different states. They then apply their model to some rat perceptual learning data in which the probability of left or right decisions (i.e., a bias) should increase or decrease across blocks of trials. They find some apparent evidence of a four state model in which left or right bias appears to be crossed with engaged or not (so engaged left, engaged right, left bias, right bias). I think this is an interesting manuscript in that it develops a useful statistical model to capture important trends in the decision making style across trials. These HMMs with transitions across decision states are conceptually easy but very difficult to fit to data. Adding to this complexity is the possibility of four different sets of GLM weights to predict choices. Finally, the authors use a nice data set to illustrate the utility of their model – not only is there left and right biases, but these are very long experiments that would almost certainly catch the rat off task from time to time (and as their model suggests, they do engage in off task behavior).

We appreciate the reviewer's thoughtful summary of our work and their recognition of the complexity involved in fitting HMMs with transition-dependent decision-making states, especially to this large dataset collected across several labs, which adds to the complexity. The reviewer highlights several key strengths of our approach, including its ability to capture distinct decision-making states and the challenge of modeling transitions across these states in a data-driven manner. We particularly appreciate the reviewer's recognition that our method is well-suited for long experimental sessions, where animals may disengage or shift strategies over time. Our model is designed to address exactly these kinds of behavioral dynamics, which would be difficult to capture using traditional, fixed-transition HMMs.

Additionally, we acknowledge the reviewer's point that incorporating different models with different sets of GLM weights adds further complexity to model fitting. However, we view this complexity as a necessary step to ensure that the transitions between decision-making states are informed by meaningful, behaviorally relevant covariates rather than being treated as a static, unchanging process. By explicitly modeling both the transitions between states (GLM-T) and decision policies within each state (GLM-O), we provide a flexible framework that allows us to infer how behavioral states evolve dynamically. This separation enables us to distinguish factors influencing long-term strategy shifts from those affecting moment-to-moment choices.

All of that said, the novelty of the research seems to be a bit low for this particular outlet in my opinion. According to the authors, this division between two separate generalized linear model structures to guide both the transition between the states and the decisions within those states is the main novelty of their research. To me, the main contribution of their modeling efforts is that they allow for two very general (literally GLM) structures to modulate the two main components of their model, one being a transition among the states and one being the decisions within those states. While this is interesting, methodologically speaking this is not a significant leap forward. Furthermore, the evidence for a need of a separate transition GLM seems quite low. Maybe it is just a scaling issue, but Figure 2a is not really convincing that having a transition model is actually necessary.

We thank the reviewer for raising these points. While HMM-based approaches to decision-making have been previously explored, our work extends these models by dedicating two distinct GLMs to separately capture within-state decision-making and between-state transitions. Specifically, we use a multinomial GLM to model state transitions, allowing the probability of switching between strategies to depend on past behavior and external covariates, and a Bernoulli GLM to model the stimulus-dependent decision process within each state. This dual-GLM architecture allows us to **fully capture the decision-making process**, simultaneously estimating the animal's decision state and determining how strategies evolve dynamically in response to changes in the environment. This framework also **improves state inference confidence** that indicates more reliable state identification, and explicitly accounts for engaged and disengaged states, which provides a richer description of real-world behavioral dynamics.

In addition, we conducted systematic **validation using synthetic data**, showing that the model only **detects dynamic transitions when they actually exist, further confirming its utility**.

This separation is particularly valuable for modeling behavior in dynamic environments, where the decision-making process is shaped not only by sensory inputs but also by the animal's evolving internal state and memory of recent experiences. Importantly, decision-making in such dynamically changing environments has not been extensively studied, and incorporating the statistics of environmental changes into the model adds **another layer of novelty**.

Furthermore, using 5-fold cross-validation, we show that this model **significantly outperforms simpler models**, such as one-state GLM-HMM (so a GLM), demonstrating the necessity of incorporating different states and two separate GLMs to fully capture the richness of the decision-making process. This empirical validation highlights that the proposed framework is **not just a conceptual extension, but a practical improvement** that provides deeper insights into the behavioral strategies employed by animals in adaptive environments.

The reviewer suggests that the need for a separate transition GLM is not well supported and that Figure 2a does not convincingly demonstrate its necessity. **To further illustrate its impact, we have conducted additional analyses comparing models with and without GLM-T and have added figures that show how inferred state probabilities change when transition probabilities are fixed versus dynamic.** These analyses clearly demonstrate that when transition probabilities are allowed to vary based on behavioral history, the model provides a better fit to the data and more interpretable latent states.

Regarding Figure 2a, we thank the reviewer for raising this point — it is indeed a scaling effect, and we have now **revised the main text** to explicitly explain how even small differences in log-likelihood per trial, when accumulated across many trials, translate into substantial improvements in overall model fit. This clarification highlights the importance of the transition GLM (GLM-T) and its contribution to capturing dynamic behavioral changes, which may not be immediately apparent due to the compressed scale shown in Figure 2a. Even a very small improvement in log-likelihood per trial can result in a much better model fit (refer to Eq. 32 in the Methods, but in this case comparing models with and without GLM-T instead). For example, for the 4-state model, the model with GLM-T is approximately 0.015 bits per trial higher than the model without GLM-T. This means that for a dataset with 400 trials, the data would be about $2^6 \approx 64$ times more probable under the GLM-T than without it. For a dataset with 4000 trials, the data would be about $2^{60} \approx 1.15 \times 10^{18}$ times more probable under the GLM-T model. We thank the reviewer for helping us clarify this point, we have added this explanation to the main text, where Figure 2a is explained, to make this clearer.

To directly address the reviewer's question about the novelty and whether the transition GLM (GLM-T) is necessary and whether it provides clear added value, we highlight several points and present both new and existing analyses, as described below.

1) First, we conducted a model comparison using two synthetic data generated (sampled) from a model without GLM-T (left side of the figure) and from a 4-state model with GLM-T (right side of the figure). As shown in the figure below (and described in more detail in our response to Reviewer 1), this comparison demonstrates that the GLM-T substantially improves model performance when the true underlying process involves trial-dependent transitions. Importantly,

when the data are generated from a model without GLM-T, the two models perform similarly, confirming that GLM-T does not overfit or artificially inflate performance when trial-by-trial modulation of transitions is unnecessary. This improvement of 0.009 bits per trial when the data has trial-dependent transitions means that for a dataset with 5000 trials, the data would be approximately 3.52×10^{13} times more probable under the GLM-T model than under the non-GLM-T model. This provides **direct evidence that the GLM-T is both useful and necessary** when state transitions are indeed dynamically influenced by external factors, as is the case in our behavioral data.

2) Second, the model with GLM-T identifies latent states with higher probabilities compared to the model without GLM-T, as shown in the figure below (left: scatter plot, right: density plot). This directly relates to the reviewer's concern about whether the transition GLM (GLM-T) is necessary. By assigning higher probabilities to the inferred states, the **GLM-T provides more confident** and more stable state estimates, showing that trial-dependent transitions improve the ability to uncover the underlying behavioral states. This provides direct evidence that **allowing transitions to depend on external covariates (such as trial history) makes the model better at tracking the animal's true internal state.**

3) Third, as shown in Fig. 4a and 4c, the model with GLM-T captures the dynamics of state switching much more effectively, reflecting the ability of trial-dependent transitions to track shifts

in behavioral strategies over time. This further demonstrates that allowing transitions to depend on external covariates enhances the model's ability to uncover meaningful temporal structure in the data.

4) Additionally, we have included a response time plot, which shows that **incorporating GLM-T leads to a greater difference** in the 90th percentile of response time between disengaged and engaged states for all animals. This enhanced separation demonstrates that the model more effectively captures behavioral dynamics and engagement-related variability in response times. For more details, please see our response to Reviewer 1 (section 4).

5) Furthermore, the below figure provides further evidence supporting the link between satiety, reward history, and disengagement, which we uncovered by incorporating GLM-T into our model. The left panel illustrates a **positive correlation between filtered reward and the probability of being in a disengaged state**, reinforcing the idea that as animals accumulate more rewards, they tend to disengage from the task. This trend aligns with previous findings suggesting that increased satiety leads to behavioral disengagement. The red trend line highlights this relationship, showing a moderate but consistent increase in disengagement probability with higher filtered reward.

The right panel further supports this link by showing the average probability of engaged and disengaged states over the course of a session, with all sessions interpolated to 100 points and averaged across all animals. Initially, animals transition from disengaged to engaged states, indicating a **warm-up period** before reaching a stable level of task engagement. However, as the session progresses, disengagement probability steadily increases while engagement decreases, particularly toward the end of the session. This pattern aligns with the interpretation that prolonged task engagement and increasing reward accumulation contribute to disengagement, likely due to satiety effects, and further supports our findings.

By incorporating **GLM-T**, our model was able to reveal these structured, trial-dependent state transitions, a novel achievement compared to traditional fixed-transition models. This approach

allows us to capture **how behavioral states evolve dynamically over time** and provides a richer understanding of the interplay between reward history, disengagement, and satiety.

6) Additional analysis of data with bias blocks (Figure S7b) and synthetic data analysis for both models, with and without GLM-T (Figure S5), demonstrate that GLM-T effectively captures the dynamics related to bias block data and the environment dynamics. This provides further evidence that incorporating GLM-T significantly enhances the model's ability to comprehensively represent behavioral dynamics.

Although the results are also interesting, it is perhaps not that surprising that they would find a model that supports the two real conditions of the experiment (left or right bias) crossed with 'engaged' or 'not engaged'.

We thank the reviewer for raising this point. While the existence of left/right bias and engagement states naturally reflects the external structure of the task, the core novelty and power of our model lies in its ability to reveal the internal structure of the animal's decision-making process — something that cannot be reduced to these simple task-defined categories. **Our approach is not limited to labeling trials as biased or engaged; rather, we explicitly assess how past choices, sensory evidence, trial history, and reward feedback dynamically influence both within-state decision policies and the transitions between states themselves.** This allows us to track the **full temporal evolution** of strategy switching, capturing subtle forms of sequential dependence and behavioral adaptation that **would be entirely invisible in a simpler model with static transitions, or in approaches that only label strategies as either engaged or disengaged.**

Importantly, **the states we identify are discovered from the data itself, not imposed by the model.** It happens that in this task, these states strongly correlate with left/right bias and engagement because these reflect real behavioral strategies used by the animals — **but the framework is fully general and could identify different states in different tasks (so not only limited to disengaged or engaged labels),** such as states driven by sensory uncertainty, motivational changes, fatigue, or exploration-exploitation trade-offs. In fact, **this flexibility is**

one of the most valuable aspects of the approach: the same model structure can adapt to uncover distinct behavioral motifs across different experimental contexts. This makes the model broadly useful and enables discovery-based behavioral analysis rather than simple classification.

Finally, while we respect the reviewer's perspective, decision-making is far more complex than a simple binary distinction between 'engaged' and 'not engaged.' It is a complex, dynamic process shaped by multiple interacting factors, including sensory information, biases, memory of past outcomes, and evolving internal motivation. By combining a flexible state-space representation with dynamic, covariate-driven transitions, **our model provides a uniquely powerful tool to study how internal decision states evolve over time, and how they respond to the statistics of the external environment.** This is particularly critical in naturalistic and adaptive behaviors, where decision-making cannot be understood as a simple, isolated process but **as a continuously evolving dialogue between internal strategies and external contingencies.**

For these reasons, we emphasize that our approach is much more than a slight refinement of left/right bias and engagement labels — it is a complete modeling framework that provides a richer and more biologically realistic view of how decisions unfold in complex, dynamic environments.

To be honest, it was a little unclear what the inputs were to the GLM observation, despite a whole section for it in 4.5. For example, they say that 'bias' was in the set of observational covariates, but it is not defined in 4.5. It is hard to understand how a variable called 'bias' could meaningfully contribute to the fits if there are already regressors for the stimulus itself and the past choice and previous stimulus.

We appreciate the reviewer's comments regarding the inputs to the GLM observation model (GLM-O) and understand that the role of bias was not clearly defined in Section 4.5. The reviewer is correct that, without a clear definition, it can be difficult to interpret how "bias" interacts with other covariates such as past choice and stimulus history. To clarify this for readers, we have revised Section 4.5 to explicitly describe bias as a separate regressor, whose role is to capture the animal's inherent left/right preference that is present even in the absence of any stimulus or trial history effects. This type of bias term (sometimes referred to as an "offset") is standard in logistic regression and GLM-based behavioral models, where it represents a baseline tendency toward one choice, independent of external influences. Importantly, this term does not "compete" with stimulus, past choice, or previous stimulus covariates — instead, it captures unexplained side preference that remains after accounting for those external factors.

Therefore, we have divided Section 4.5 into two subsections to improve clarity and added the following explanation to explicitly describe bias as a separate regressor:

"Finally, "bias" is a covariate that captures the animal's inherent tendency to choose left or right, representing an internal preference or offset that exists independently of

external factors such as the stimulus or choice history, and its value was set to 1 for all trials.”

In any case, I would worry that they are finding evidence for a four state model simply because the models that explain decisions are far too simplistic. It is possible that a more detailed model of the decision itself might imply that there are actually fewer states. For example, potentially even having a regressor for ‘number of trials performed in this block’ might soak up a lot of within-state variability to remove the need for two of those four states. Because questions like this were not really fully examined, the results surrounding the transition GLM’s contribution are pretty minimal. However, even if they were significant, it is not clear that an incremental methodological advancement in a well established field of HMMing makes sense at this particular outlet.

Thank you for raising this point. Our decision (observation) model design is aligned with similar literature [31-34], and we respectfully do not agree with the reviewer that it is overly simplified. We carefully selected reasonable covariates to ensure the model appropriately captures behavioral complexity. We believe this model is well-suited to capturing the richness of decision-making, particularly in adaptive environments where behavior reflects not only immediate sensory inputs but also sequential history and environmental context.

Furthermore, our model selection process itself was guided by rigorous 5-fold cross-validation, ensuring that the choice of the model was based on empirical evidence rather than subjective preference. In our case, the GLM observation model (GLM-O) already incorporates stimulus, past choice, past stimulus, and bias, explicitly accounting for both sensory-driven decisions and sequential effects like choice and reward history — factors known to strongly influence behavior in perceptual decision-making tasks. **This level of detail in the observation model reduces the risk of spurious state discovery caused by missing within-state structure.**

Moreover, the four states identified by our model **are not arbitrary categories introduced to artificially improve fit, but rather emerge consistently across animals and sessions, with clear behavioral interpretation.** Importantly, the transitions between these states align with task structure (block switching) and show systematic modulation by trial history covariates such as past reward and choice. **This coherence between inferred states, task design, and behavioral interpretation provides strong evidence that the four-state solution reflects meaningful latent cognitive strategies, rather than compensating for a weak within-state model.**

Finally, we respectfully disagree with the characterization of this work as an "incremental methodological advance." While GLM-HMMs have been used before, the combination of dynamic, covariate-dependent state transitions (GLM-T) with a behaviorally rich observation model (GLM-O) is novel in its ability to capture how animals dynamically adapt strategies in non-stationary environments — something that is essential for studying behavior in tasks with evolving contingencies. Also, this framework is broadly applicable beyond the specific task studied here and **offers a flexible, interpretable tool for understanding adaptive behavior across diverse experimental paradigms.** This conceptual and practical advance, **supported**

by both synthetic and real data, goes well beyond fitting a better curve and provides novel insights into how animals dynamically structure their behavior.

In summary, the four-state structure reflects a biologically interpretable hierarchy of behavioral modes, **supported by the data, the task design, and established principles of decision-making behavior — not a byproduct of an overly simplistic within-state model.** We appreciate the opportunity to explain this key aspect of our approach. But we appreciate the reviewer’s point that adding more covariates could provide additional clarity and enhance the model’s interpretability (in addition to GLM-T and GLM-O). We have included a few lines in the future work section to address this.

Finally, we have organized our response into three sections based on the reviewer’s concerns:

1. The reviewer is concerned that the four-state structure might result from an insufficiently detailed observation model rather than reflecting genuine behavioral states. The four-state model is empirically justified and emerges naturally from the data rather than being imposed due to a limited GLM-O. Additionally, as mentioned, a **5-fold cross-validation** was performed, ensuring that the model selection was based on empirical evidence rather than subjective preference. Another important point is that incorporating GLM-T allows us to gain deeper insights into **state transition patterns**—such as the effects of different covariates, their historical influence on transition probabilities, and how states evolve—**without manipulating the observation model.**

a) Figures comparing state occupancy patterns in **4-state models with and without GLM-T, for both real and synthetic data, demonstrate that with GLM-T, states align meaningfully with task biases, whereas without it, state occupancies lack structure.** To further address the reviewer’s question, we plotted the fractional occupancy of the four discrete states in the model without GLM-T for all IBL data (similar to Fig. 4c). This analysis was conducted across all trials, covering both right-biased and left-biased blocks, with results displayed separately for each block type. As shown in the plot, the background stimulus bias was not captured by the model without GLM-T.

In addition, please see the figure below, which is a supplementary figure from the paper employing synthetic data. This figure illustrates the fractional occupancy of different states for models with and without GLM-T, separated by left-biased and right-biased task conditions. The

comparison shows that in a **4-state model** when **GLM-T is included (panel a)**, the **state occupancy distributions adapt more distinctly to task biases**, whereas without GLM-T (panel b), the state occupancies appear without being affected by data bias blocks. This further supports that the inclusion of trial-dependent transitions allows the model to capture dynamic shifts in behavioral states.

b) Additionally, posterior state probabilities across trials indicate that states inferred under GLM-T are assigned higher probabilities (Figure S9), demonstrating that the **states are well-defined and not arbitrary**. Figures comparing scatter plots of inferred state probabilities show that states remain well-separated and confidently assigned under GLM-T, **which would not be the case if the model were unnecessarily splitting states**.

Furthermore, as mentioned in the previous response, the state probability plot comparing the model with GLM-T to the model without GLM-T shows that inferred states under GLM-T exhibit higher certainty and structure. **This suggests that the model is detecting meaningful behavioral states rather than artificially splitting states**.

2. The reviewer suggests that adding a within-state regressor could absorb variability that currently leads to the four-state model, potentially making fewer states sufficient. The four-state model captures **discrete behavioral shifts** that a continuous regressor like "number of trials in a block" could not. To illustrate this, we plotted the transition probabilities for both models—with and without GLM-T—to highlight their differences over time. The plots show that incorporating GLM-T allows **transition probabilities to vary dynamically from trial to trial**, adapting to different states and conditions in a way **that a simple within-state regressor could not capture**. In contrast, without GLM-T, transition probabilities remain constant, failing to reflect behavioral changes. This comparison underscores that the extended model effectively captures adaptive state transitions, distinguishing it from the standard GLM-HMM, which assumes fixed transitions that do not evolve based on trial history or task structure.

Additionally, the response time plot referenced in the previous response further supports this by showing better state separation between disengagement and engagement when GLM-T is included. Collectively, these analyses demonstrate that without GLM-T, state transitions lack structure, **providing evidence that the four-state model with GLM-T captures genuine behavioral shifts rather than compensating for missing within-state regressors.**

Finally, please consider that GLM-T **estimates transition probabilities between latent states**, meaning it determines how an animal switches between different behavioral modes. These transitions are influenced by multiple covariates, including past choices, rewards, and task structure, allowing for adaptive, history-dependent behavioral shifts. In contrast, "number of trials in a block" is a **continuous regressor** that only tracks the progression of time within a block and affects decision weights within a single state. It assumes that changes in behavior occur gradually within a state, rather than through distinct transitions between states. **Thus, while "number of trials in a block" might account for a small portion of choice behavior changes within a given state, it fails to capture the actual switching between states, which is the core function of GLM-T. Our model is significantly more powerful, as it not only tracks within-state decision dynamics but also accurately models trial-dependent state transitions, allowing for a much richer and behaviorally meaningful representation of adaptive decision-making.**

3. The reviewer questions whether adding a separate transition GLM is a meaningful innovation or just a minor refinement of existing HMM approaches. GLM-T provides a fundamentally improved framework for capturing behavioral state transitions, validated through both synthetic data and real experimental results. Figures from the model comparison using synthetic data demonstrate that GLM-T improves performance only when trial-dependent transitions exist, confirming its necessity. Additionally, log-likelihood improvement analyses show that even small per-trial gains translate into substantial benefits in large datasets. This was previously mentioned, but we reiterate it here for clarity. As shown in the figure below, an improvement of 0.009 bits per trial when the data contains trial-dependent transitions means that for a dataset with 5000 trials, the data would be approximately 3.52×10^{13} times more probable under the GLM-T model than under the non-GLM-T model.

This analysis validates that GLM-T improves model performance only when trial-dependent transitions are present, demonstrating that it **captures meaningful behavioral dynamics rather than serving as a minor refinement**. This suggests that transitions between latent states are not merely stochastic but are systematically influenced by past experiences, **reinforcing the necessity of modeling them with a flexible, input-driven approach**.

Reviewer #3 (Remarks to the Author):

The authors investigated behavioral strategies used by mice performing a binary decision-making task with a time-varying distribution over correct choices. To do so, they fit two sets of GLMs: one that predicts observed choices as a function of sensory inputs, task covariates, and latent state, and another that predicts transition probabilities to each latent state. Crucially, this setup allows the transition probabilities to vary over time, in contrast with traditional HMMs, and thus is a sensible choice for analyzing behavior in a non-stationary environment. In sum, they find four interpretable states that govern behavior during this task, which are broken down into “engaged” states, in which animals appear to rely on the observed stimulus, and “biased” states, in which animal’s choices seem heavily biased towards one direction regardless of stimulus. They characterize various task-relevant factors that influence

transitions between these states, and provide a detailed account of how transition rates are modulated over time by the underlying evolving task structure.

Strengths: The authors do an excellent job demonstrating that the behavioral data is well-explained by their proposed four-state model, and their interpretation of these states is solidly supported by breakdowns of GLM weights and psychometric curves.

We sincerely appreciate the reviewer's thoughtful feedback and recognition of our modeling approach. Your comments on the interpretability of the identified states and our framework's ability to capture behavior in a non-stationary environment are highly encouraging.

Concerns/Questions:

The relationship between this work and the cited "Extracting the dynamics of behavior in sensory decision-making experiments" (Roy et al., 2021, Neuron) should be explicitly discussed. In particular, both make inferences about animal behavior from the same dataset, and both employ a fitting method that permits a time-varying observation model. The authors should make an effort to underscore what findings are uniquely highlighted by their GLM-HMM approach that couldn't be captured by the model used there.

Thank you for this point. While both our study and Roy et al. (2021) analyze behavioral strategies in decision-making tasks using time-varying models for observation, the scope, methodology, and focus of the two approaches differ significantly. Roy et al. (2021) focused on learning dynamics, applying PsyTrack to track smooth, trial-by-trial fluctuations in decision-making weights, such as sensory evidence weighting. Their method is particularly well-suited for studying how decision strategies evolve as animals **learn the task** and adapt to changing task statistics. PsyTrack continuously adjusts the model parameters over time, **capturing gradual learning-driven changes in decision-making behavior**. They used the training data to model learning.

In contrast, our study examines **behavior after learning** has stabilized, focusing on how trained mice dynamically **make decisions or transition between distinct latent behavioral states** during task performance. Rather than modeling continuous weight updates, we apply a GLM-HMM framework that infers discrete states (e.g., engaged vs. biased states) and analyzes how transitions between these states are influenced by sensory inputs, past choices, and task structure. This discrete-state representation allows us to investigate abrupt strategy shifts, providing insight into how trained animals dynamically engage with or disengage from the task. Also, prior work (Zoe et al., 2022, Nature Neuroscience) has shown that **mice use discrete, rather than continuously evolving, decision-making states (GLM-HMM achieved substantially higher test log-likelihood than the PsyTrack model)**, further motivating our use of a GLM-HMM framework instead of a smooth-tracking approach like PsyTrack.

In terms of methodology, Nick Roy's model employs a **dynamic Bernoulli GLM, without any HMM, where psychophysical weights evolve smoothly over time** to track how decision-making strategies change from trial to trial. This approach allows the model to capture gradual adaptation by adjusting the influence of task variables (e.g., sensory stimuli, biases,

choice history) on choices. However, it lacks an explicit latent state structure, meaning it does not model discrete shifts in behavioral strategies that might arise due to factors such as engagement level or block transitions. In contrast, our method integrates a GLM-HMM framework, which introduces latent behavioral states and allows for history-dependent state transitions. Our model **consists of an HMM and two separate Bernoulli and multinomial GLMs**: GLM-O (observation model), which governs within-state decision-making based on task variables, and GLM-T (transition model), which predicts how and when animals switch between different decision-making states based on past choices, rewards, and other covariates. This distinction is crucial, as it enables our model to capture both gradual within-state adaptation and discrete shifts between cognitive states, which a standard dynamic GLM like Roy’s cannot. Additionally, our framework allows for trial-dependent transition probabilities, making it more suitable for analyzing adaptive, non-stationary behaviors where animals dynamically shift between strategies in response to the task structure.

This distinction enables us to characterize global shifts in decision-making behavior, offering a novel framework for understanding structured strategy adaptation in well-trained animals—something that may not be fully captured by a smooth-tracking approach like PsyTrack. Importantly, our approach allows for discovery of latent behavioral states directly from data, rather than only tracking pre-specified weights. We have clarified these differences in the manuscript (in introduction with orange color) and appreciate the opportunity to better articulate the unique contributions of our approach. We have provided a comparison table as below to address the different features of both methods more clearly.

Feature	PsyTrack (Roy et al., 2021)	Our Model
Main Goal	Track continuous learning dynamics over training	Capture discrete behavioral states and dynamic transitions during stable behavior after learning
Type of Model	Dynamic GLM with time-varying weights	Hidden Markov Model with Bernoulli GLM for within-state choices and multinomial GLM for transitions
Focus	Evolution of psychophysical weights across trials during training	State switching (strategy shifts) and within-state choice policies during steady-state behavior
What is modeled	Continuous weight trajectories that evolve smoothly across trials	Discrete latent states (e.g., engaged, biased) and probabilistic transitions between them
State Transitions	No explicit states, so no transitions	Explicit state transitions modeled via a multinomial GLM (GLM-T) informed by past trials
Trial history effects	Captured indirectly through smoothly evolving weights	Directly modeled as covariates in GLM-T (e.g., past reward, past choice)
Flexibility	Can track arbitrary weights if included as covariates	Can discover distinct behavioral states, each with its own decision policy, allowing for richer interpretation
Application focus	Ideal for early learning and tasks with gradual adaptation	Ideal for adaptive decision-making in stable, dynamic environments after learning completes

In order to address the reviewer's question, we present a few additional analyses here:

1. GLM-HMM Captures Discrete Behavioral States, Unlike Psytrack's Continuous Weight Evolution: A key distinction between our GLM-HMM approach and Nick Roy's Psytrack-based model is that our framework explicitly infers discrete behavioral states, rather than assuming decision-making changes occur as a continuous weight evolution. To better address this, we examined the fractional occupancy of the four discrete states in the model without GLM-T for all IBL data, similar to Fig. 4c. This analysis was conducted across all trials, covering both right-biased and left-biased blocks, and results were displayed separately for each block type.

As seen in the plot, the background stimulus bias was not captured by the model without GLM-T, demonstrating that without explicit state transitions, the model **fails to correctly align behavioral states with task structure**. This reveals a fundamental limitation of Psytrack: much like a GLM-HMM without GLM-T, it tracks gradual changes in decision weights but fails to infer structured latent states that represent discrete shifts in decision-making strategies. In contrast, **GLM-HMM provides a richer representation by capturing discrete cognitive states and their transitions, which are essential for understanding dynamic behavioral patterns in non-stationary environments.**

2. GLM-T Captures Trial-Dependent Transitions That Psytrack and Standard GLMs Cannot: GLM-T provides a fundamentally improved framework for modeling behavioral state transitions, as validated through both synthetic data and real experimental results. Unlike Psytrack, which assumes gradual, continuous changes in decision-making behavior, GLM-T **explicitly models trial-dependent state transitions, allowing it to capture discrete shifts between behavioral strategies rather than smooth weight updates.**

We compared model performance using synthetic datasets generated from two different sources: one sampled from a model without GLM-T (left side of the figure) and another from a 4-state model incorporating GLM-T (right side of the figure). As illustrated in the figure below (and further detailed in our response to Reviewer 1), this analysis confirms that GLM-T provides a significant advantage when trial-dependent transitions are present in the underlying process. However, when the data originate from a model without GLM-T, both models perform comparably, demonstrating that GLM-T does not introduce overfitting or artificially enhance performance in cases where trial-by-trial transition modulation is unnecessary.

As previously discussed, an improvement of 0.009 bits per trial when the data contains trial-dependent transitions means that for a dataset with 5000 trials, the data would be approximately 3.52×10^{13} times more probable under the GLM-T model than under the non-GLM-T model. In contrast, Psytrack does not model **state transitions at all**, reinforcing that our approach offers a significant methodological advancement.

3. GLM-HMM Provides More Confident and Structured State Estimates: We compared models with and without GLM-T, and the results show that the model with GLM-T assigns higher probabilities to inferred states, as seen in the figure below (below scatter plot). This indicates that GLM-T provides more confident and stable state estimates, reinforcing its ability to uncover meaningful behavioral states. Our model captures structured, discrete state transitions better than models without a transition component. **In contrast, models without GLM-T, similar to Psytrack, lack a mechanism to explicitly model state-to-state transitions and instead assume that behavior changes gradually and continuously.** If a purely continuous process were sufficient, we would not observe such high-probability state assignments in the model with GLM-T.

Additionally, the response time analysis (Fig. 7a) further supports the behavioral relevance of these inferred states, showing that the separation between engaged and disengaged states is clearer when GLM-T is included. This reinforces the idea that decision-making is not just a continuous adaptation but involves distinct transitions between cognitive strategies.

Together, these findings demonstrate that GLM-HMM with GLM-T provides a more structured and confident inference of behavioral states compared to models without an explicit transition component, such as Psytrack.

I am somewhat puzzled by why state transition probabilities are modeled by the transition GLM in the way they are. If I am understanding correctly, only the probabilities of transitioning to a particular latent state are parameterized, regardless of starting latent state, for a total of K transition probabilities (this interpretation is consistent with the exposition in the text as well as Fig. 5). This is quite a restrictive choice compared to standard HMMs, which parameterize all K^2 pairs of from-and-to. In this case, I would expect that transitions “Engaged R → Biased R” would be significantly more probable than “Engaged L → Biased R”. Indeed, other works that use time-varying transition probabilities, such as the cited work Calhoun et al. 2019, fit a separate GLM for the transition probabilities out of each latent state. I would like to see that the conclusions drawn are not overly dependent on this particular modeling choice.

We appreciate the opportunity to clarify this point. Our transition model does not impose a single set of transition probabilities across all states. Instead, **we fit a separate multinomial GLM (GLM-T) for each latent state (see sections 4.3 and 4.4 of Methods), similar to Calhoun et al. (2019)**. However, a key difference in our approach is that the covariates for GLM-T are independent from those used in the observation GLM (GLM-O), allowing to distinguish between factors driving within-state choices and those influencing state transitions.

This framework ensures that transition probabilities remain state-dependent, meaning the probability of entering a given state varies based on both the current latent state and external covariates, such as past rewards, past choices, and session warm-up effects. By modeling transitions with a multinomial GLM rather than a fixed $K \times K$ transition matrix, our approach naturally captures asymmetries in state transitions, enabling behaviorally meaningful shifts between engaged and biased states. Furthermore, this design allows us to better model decision-making in non-stationary environments, where animals must dynamically adjust their strategies in response to changing reward contingencies. This level of flexibility is not present in standard HMMs, which assume fixed transition probabilities over time.

While directions for future work are discussed thoroughly, interpretations and caveats of the presented findings seemed a tad scarce in the discussion. For example, the authors allude to occupancy of disengaged states being related to satiety. The authors could expand on this, possibly by commenting on the robustness of this conclusion, especially when considering that the delineation appears less clear for the 5 state model as shown in Fig. S2b.

We appreciate this thoughtful point and agree that further elaboration on the interpretation and robustness of our findings would strengthen the discussion. Our results for 4-state show that past rewards are a predictor of transitions to disengaged states, aligning with the idea that satiety may contribute to disengagement. Importantly, while satiety provides a compelling explanation, disengagement may also incorporate other factors such as temporary reductions in task engagement or shifts in decision strategy. We will clarify these points especially for the

5-state model in the discussion to highlight the robustness of our findings while acknowledging the broader behavioral context in which disengagement occurs.

Also, to address the reviewer's question regarding the interpretation of disengaged state occupancy and its relationship to satiety, we provide additional analysis supporting this link. The left panel of the figure below demonstrates a positive correlation between accumulated reward (filtered reward) and the probability of being in a disengaged state. We found a statistically significant positive moderate correlation between the filtered reward rate and the probability of disengagement ($r \approx 0.30$, $p < 0.001$), where $r = 0.30$ (Pearson correlation coefficient) indicates a moderate positive correlation, and $p < 0.001$ (p-value) confirms that this relationship is highly unlikely to be due to chance. This trend suggests that as animals receive more rewards over time, they are increasingly likely to disengage from the task, aligning with the hypothesis that **satiety drives disengagement**. This result reinforces the robustness of our interpretation, as it is consistent with previous findings that reward accumulation influences behavioral state transitions.

The right panel further supports this by illustrating the average probability of engaged and disengaged states over the course of a session, with all sessions interpolated to 100 points and averaged across animals. Initially, animals exhibit a transition from disengaged to engaged states, reflecting an early adaptation or warm-up period before stable engagement. However, as the session progresses, the likelihood of disengagement increases, particularly toward the end of the session, while engagement probability decreases. This pattern is consistent with the idea that **extended task participation and accumulating rewards lead to disengagement, likely due to satiety effects**.

Minor/miscellaneous:

Is there a relationship between the model's predicted probability of the animal being in a biased/disengaged state during some time window and the number of incorrect choices observed? The authors state that performance remains high even in the biased states because of imbalanced prior probabilities, but I don't see this quantified anywhere. Reporting a

prevalence-independent performance metric like balanced accuracy could also be insightful here.

We thank the reviewer for their insightful comment. To address this, we present the normalized error rate across all sessions of all animals for both engaged and disengaged states. While the error rate in the disengaged states is slightly higher than engaged states, it remains relatively low overall, demonstrating that the model continues to perform well even in these periods. The use of a normalized error rate ensures that differences are not driven by variability in trial numbers, providing a clearer comparison. This analysis directly reflects the relationship between an animal being in a biased/disengaged state during the experiment and the rate of incorrect choices observed. Our results support the explanation that performance remains high in the disengaged state due to imbalanced prior probabilities, as errors do not increase drastically and provides meaningful quantification of state-dependent performance, reinforcing our interpretation.

Reviewer #4 (Remarks to the Author):

Thank you for your contribution to the review process. We appreciate the thoughtful feedback and the opportunity to improve our manuscript through this collaborative review.

Thank you again for your comments and helpful suggestions.

Sincerely,

Zeinab Mohammadi, Zoe C Ashwood, The International Brain Laboratory, and Jonathan W
Pillow

Dear Reviewers,

Thank you once again for your thoughtful and constructive feedback on our manuscript. We appreciate the opportunity to address this second round of comments. Your insights have helped us further refine the clarity and depth of the work. Below, we provide point-by-point responses and describe the corresponding changes made to the manuscript.

REVIEWER COMMENTS

Reviewer #1 (Remarks to the Author):

We would like to thank the authors for thoughtfully and thoroughly addressing our concerns. Below, we have outlined a few minor final points.

Thank you for your careful review and constructive feedback. We have addressed the remaining points as outlined below and made the corresponding edits to the manuscript and figures.

Rebuttal point 3:

"... This result highlights that our model, which integrates both GLM-T and GLM-O, governs how behavioral states evolve over time while capturing choice behavior patterns, whereas GLM-O alone defines within-state choice behavior. The consistency of GLM-O weights across models further supports the idea that the transition process (GLM-T) does not interfere with moment-to-moment decision weightings but instead regulates long-term behavioral shifts. We have added this plot to the supplementary materials."

Thank you for running this additional analysis. We now see that the GLM-O weights do not really differ between the models with or without the GLM-T component. We were wondering which part of the main text (if any) refers to these results. It might be worth explaining this point in a little more detail, because it was at first counterintuitive that the two weights should not influence each other.

Thank you for this suggestion. We agree that the relationship between the GLM-T and GLM-O components warrants clearer explanation in the main text. As noted in the rebuttal, the consistency of GLM-O weights across models with and without GLM-T indicates that GLM-T governs transitions between latent behavioral states without altering state identities themselves, as captured by GLM-O.

To clarify this, we have now added a sentence in the Results section (Lines 430–433):
"Furthermore, GLM-O weights were similar across models with and without the GLM-T component (Supplementary Fig. S7a). This shows that the GLM-T regulates transitions

between behavioral states without substantially modifying the states themselves, as described by the GLM-O."

Rebuttal point 4:

Thank you for following up with a comprehensive set of analyses. The result suggests that including GLM-T component leads to higher confidence in state assignments with distinct predictions for behavior such as RT and switching. The new figures were overall clear, with the exception of Fig.9c: the plots were too cluttered and the analysis here seems to rely on qualitative judgment. Could this be replaced, for example, with the entropy of the $P(\text{state})$ or showing just for the maximum probability instead, to avoid redundancy with panels (a)-(b)? Also we assume this is for an example session. The figure is also missing the legend for the colors of the plots.

We thank the reviewer for the helpful suggestion. We have revised Fig. S9c to reduce visual clutter by replacing the original plot with a plot of the maximum posterior probability over time, which more directly reflects confidence in state assignments. As shown in the updated panel, Fig. S9c now presents a direct comparison between the models with and without GLM-T. The model with GLM-T (solid purple line) mostly assigns higher confidence to latent state inferences than the model without GLM-T (dashed line), with confidence quantified as the maximum posterior probability per trial. This change avoids redundancy with panels (a)–(b), which already depict the full state probability profiles. We have also clarified in the figure caption that the plot is based on an example session, added a legend for state colors, and updated the caption accordingly.

Rebuttal point 5:

Thank you for testing this idea. We think the results are clear.

We thank the reviewers for their positive feedback. We are glad the results are clear and satisfactory.

Rebuttal point 6:

Thank you for providing these extensive simulation results. The model recovery results and Fig.S10 were clear in demonstrating the utility of the transition model. We would just recommend clarifying the simulation conditions (e.g., parameters) for generating the synthetic data from two models in Methods, if this has not already been done.

We thank the reviewers for their helpful feedback and are glad the simulation and model recovery results were clear. In response to your suggestion, we have now added a description of the simulation conditions to generate synthetic data from the two models, as described in the Methods section titled "Synthetic Data" (Section 4.9).

Rebuttal point 7:

Thank you for the additional explanation.

We appreciate the reviewer's feedback and are glad the additional explanation was helpful.

Reviewer #2 (Remarks to the Author):

I don't really have any further comments. Although I appreciate the authors' efforts, the model they use here is more of a statistical approach (through the use of very general GLMs) and lacks a lot of theoretical interest, which is the type of approach I usually gravitate toward. The authors did a lot to justify their position, and while I appreciate their efforts, the manuscript remains in a similar state as the first version although some of the additional analyses are slightly more convincing statistically, it is still not clear to me whether the results are sort of guaranteed from the structure of the task. We continue to disagree about the novelty of their approach for this particular high-impact journal, but it is not up for me to decide so hopefully the AE can make a decision here.

We thank the reviewer for their comments and for acknowledging the added weight of the additional statistical analyses. We recognize that our statistical modeling approach differs from the kinds of theoretical models the reviewer might prefer, but we respectfully disagree with the assertion that the results are guaranteed from the structure of the task. For example, we do not feel it is obvious that mice would exhibit both engaged and disengaged states with left and right biases, respectively. It might have been that their engaged states reflected a sensitivity to the left and right blocks, but that they had only a single disengaged state with no bias to either side. Similarly, we felt it *could* have come out that the factors driving transitions into the engaged states could have been an increase (rather than a decrease) in recent reward. (Our colleagues who train animals have told us they sometimes try to drive an animal back into an engaged state by giving them an extra large reward).

More generally, the fact that we can recover stable, interpretable states across animals — along with consistent within-state decision weights and behaviorally relevant transition dynamics — represents a novel finding that we feel has not been predicted (to our knowledge) from previous normative accounts of behavior. We hope the editor will consider this broader theoretical contribution, especially given the growing recognition of statistical approaches in modeling for decision-making. (Although we are very grateful to the reviewer for their respectful expression of their difference in opinion about this issue!).

Reviewer #3 (Remarks to the Author):

I thank the authors for their additional analyses and for the expanded discussions in response to my points. I have one remaining comment.

Re: Response to “I am somewhat puzzled by why state transition probabilities are modeled by the transition GLM in...”

If a separate multinomial GLM is indeed being fit for each latent state, I would interpret that as meaning a separate set of K weight vectors is used for modeling the transition probabilities of leaving each state. However, Based on equation (2) and lines 90-94, as well as equation (12) and lines 607-611, the multinomial GLM used for state transitions parameterizes probabilities of transitioning to states with just one set of K weight vectors. What is still not clear here is how these probabilities take into account the starting state from which the state transition is occurring. Is starting state information somehow implicitly embedded in \mathbf{x}_t^{tr} ? If not, then the problem of indistinguishability of $P(\text{Engaged R} \rightarrow \text{Biased R})$ vs $P(\text{Engaged L} \rightarrow \text{Biased R})$ remains.

This point remains my biggest concern. If K separate multinomial GLMs are being fit for transition probabilities, the quoted equations/text should be updated to clarify this.

We thank the reviewer for this comment, and we apologize for the confusion over this point. There was a miscommunication between the authors during the previous round of review, and it appears that our previous response about the transition GLMs was incorrect. We sincerely apologize for this oversight!

The reviewer is correct that we used only K linear weight vectors for the GLM-T model, one for the *incoming* transitions to each state. However, the probability of transition also depended on an additive constant, one for each pair of states. Thus, the probability of transitioning from state i to state j at time t , given covariates \mathbf{x}_t^{tr} , is modeled as:

$$p(z_t = j \mid z_{t-1} = i, \mathbf{x}_t^{\text{tr}}) \propto \exp \left(B_{ij} + \mathbf{w}_j^{\text{tr}\top} \mathbf{x}_t^{\text{tr}} \right)$$

This means:

- B_{ij} is a learned baseline transition logit from state i to j (as in a standard HMM).
- \mathbf{w}_j^{tr} is a vector of covariate weights that modulate all incoming transitions to state j .

We have added this as Equation (14) in the Methods section and Equation (3) in the Results section in order to clarify the model structure.

If K denotes the number of states and d denotes the number of covariates, then the model contains K^2 parameters for B_{ij} , and $K \times d$ parameters for the transition weight vectors \mathbf{w}_j^{tr} . This leads to a total of $K^2 + Kd$ transition parameters. So considering $K=4$ states and $\text{dim}(\mathbf{x}) = 6$ (i.e., $d = 6$) transition covariates, we have **40** transition parameters in total. (The number of

degrees of freedom is actually $K(K - 1) + (K - 1)d$, due to the fact that the probabilities for each starting state i must sum to 1.)

By contrast, a standard HMM contains K^2 parameters, so only **16** parameters here for $K=4$ (or $K(K-1)$ degrees of freedom since every row of the transition matrix must sum to 1). Therefore, our GLM-based transition model is strictly more expressive and flexible – the standard HMM is a special case of our model when all transition GLM weights are zero.

We note that other works such as Calhoun et al. (2019) used GLMs with separate weights for each transition (i, j) which allows for greater flexibility, but also requires more data and is more susceptible to over-fitting. Our formulation thus offers a middle ground: it allows covariate-driven transitions without requiring separate filters per origin state. Even though w_j^{tr} does not depend on i , the presence of B_{ij} ensures that the transition probability *does* depend on the previous state.

However, to address the reviewer’s original concern and to validate this modeling choice, we performed a new analysis of the state transitions inferred by our model. In particular, we computed the conditional probability of all transitions to a state given the previous state (e.g., to assess whether it is more likely to transition to the **Biased-R** state from the **Engaged-R** state than from the **Biased-L** state). This analysis, shown below, revealed that the transition probabilities into a given state are **similar across different preceding states**. This suggests that the model’s transition dynamics were not heavily influenced by the specific prior state, thereby suggesting that using different transition filters for each starting state would not substantially improve the model.

For IBL data:

We have included this analysis as Supplementary **Figure S11** in the revised manuscript, along with an explanation in the main text (lines 433–435) as below:

“Fig. S11 shows that transition probabilities into a given state are nearly identical across different preceding states, for IBL data. This confirms that our key conclusions are not sensitive to this aspect of the model.”

We also included this in the Results section for added clarity (lines 97 to 106):

“To limit model complexity while still capturing covariate-driven transitions, we used one transition weight vector per destination state (w_j^{tr}), along with a baseline term for each state pair (B_{ij}). Therefore, the probability of transitioning from state i to state j at time t is modeled as:

$$p(z_t = j \mid z_{t-1} = i, \mathbf{x}_t^{\text{tr}}) \propto \exp \left(B_{ij} + \mathbf{w}_j^{\text{tr}\top} \mathbf{x}_t^{\text{tr}} \right)$$

This structure allows transitions to depend on both the previous state—through B_{ij} , a learned baseline transition logit from state i to j —and the upcoming state, enabling covariate-driven transitions without requiring separate filters for each origin state. We verified that transitions into a given state did not differ significantly depending on the preceding state, supporting this modeling choice (Supplementary Figure S11).”

And similarly we explained it in the method section (Equation (14)).

Reviewer #4 (Remarks to the Author):

We thank Reviewer #4 for their contribution to the review process as part of the Nature Communications initiative to support Early Career Researchers. We appreciate the time and effort they dedicated to co-reviewing our manuscript.

A point-by-point response to the reviewers' comments

We are grateful to the Editor and Reviewers for the time and care devoted to our manuscript. Your comments helped us sharpen the presentation and improve clarity. We appreciate the constructive feedback throughout the review process.

Reviewer #1 (Remarks to the Author):

The authors have adequately addressed our questions and comments, and we have no further remarks.

Response: We thank the reviewer for this positive assessment.

Reviewer #1 (Remarks on code availability):

I did not test the code but the it seems well documented.

Response: Thank you. The public repository includes a concise README with step-by-step instructions, versioned dependencies (Conda environment), and figure-level scripts to reproduce results.

Reviewer #3 (Remarks to the Author):

Thank you for your response. I don't have any more comments or questions.

Response: We thank the reviewer for their careful evaluation and are pleased that no further issues remain.